# TRAIL: NEAR-OPTIMAL IMITATION LEARNING WITH SUBOPTIMAL DATA

**Mengjiao Yang**
UC Berkeley, Google Brain
`sherryy@google.com`

**Sergey Levine**
UC Berkeley, Google Brain

**Ofir Nachum**
Google Brain

## ABSTRACT

The aim in imitation learning is to learn effective policies by utilizing near-optimal expert demonstrations. However, high-quality demonstrations from human experts can be expensive to obtain in large number. On the other hand, it is often much easier to obtain large quantities of suboptimal or task-agnostic trajectories, which are not useful for direct imitation, but can nevertheless provide insight into the dynamical structure of the environment, showing what *could* be done in the environment even if not what *should* be done. We ask the question, is it possible to utilize such suboptimal offline datasets to facilitate *provably* improved downstream imitation learning? In this work, we answer this question affirmatively and present training objectives that use offline datasets to learn a *factored* transition model whose structure enables the extraction of a *latent action space*. Our theoretical analysis shows that the learned latent action space can boost the sample-efficiency of downstream imitation learning, effectively reducing the need for large near-optimal expert datasets through the use of auxiliary non-expert data. To learn the latent action space in practice, we propose TRAIL (Transition-Reparametrized Actions for Imitation Learning), an algorithm that learns an energy-based transition model contrastively, and uses the transition model to reparametrize the action space for sample-efficient imitation learning. We evaluate the practicality of our objective through experiments on a set of navigation and locomotion tasks. Our results verify the benefits suggested by our theory and show that TRAIL is able to improve baseline imitation learning by up to 4x in performance.

## 1 INTRODUCTION

Imitation learning uses expert demonstration data to learn sequential decision making policies (Schaal, 1999). Such demonstrations, often produced by human experts, can be costly to obtain in large number. On the other hand, practical application domains, such as recommendation (Afsar et al., 2021) and dialogue (Jiang et al., 2021) systems, provide large quantities of offline data generated by suboptimal agents. Since the offline data is suboptimal in performance, using it directly for imitation learning is infeasible. While some prior works have proposed using suboptimal offline data for offline reinforcement learning (RL) (Kumar et al., 2019; Wu et al., 2019; Levine et al., 2020), this would require reward information, which may be unavailable or infeasible to compute from suboptimal data (Abbeel & Ng, 2004). Nevertheless, conceptually, suboptimal offline datasets should contain useful information about the environment, if only we could distill that information into a useful form that can aid downstream imitation learning.

One approach to leveraging suboptimal offline datasets is to use the offline data to extract a lower-dimensional *latent action space*, and then perform imitation learning on an expert dataset using this latent action space. If the latent action space is learned properly, one may hope that performing imitation learning in the latent space can reduce the need for large quantities of expert data. While a number of prior works have studied similar approaches in the context of hierarchical imitation and RL setting (Parr & Russell, 1998; Dietterich et al., 1998; Sutton et al., 1999; Kulkarni et al., 2016; Vezhnevets et al., 2017; Nachum et al., 2018a; Ajay et al., 2020; Pertsch et al., 2020; Hakhamaneshi et al., 2021), such methods typically focus on the theoretical and practical benefits of *temporal abstraction* by extracting temporally extended skills from data or experience. That is, the main benefit of these approaches is that the latent action space operates at a lower temporal frequency than the original environment action space. We instead focus directly on the question of *action representation*: instead of learning skills that provide for temporal abstraction, we aim to directly reparameterize the action space in a way that provides for more sample-efficient downstream

Figure 1: The TRAIL framework. Pretraining learns a factored transition model $\mathcal{T}_Z \circ \phi$ and an action decoder $\pi_\alpha$ on $\mathcal{D}^{\text{off}}$. Downstream imitation learns a latent policy $\pi_Z$ on $\mathcal{D}^{\pi_*}$ with expert actions reparametrized by $\phi$. During inference, $\pi_Z$ and $\pi_\alpha$ are combined to sample an action.

imitation without the need to reduce control frequency. Unlike learning temporal abstractions, action reparamtrization does not have to rely on any hierarchical structures in the offline data, and can therefore utilize highly suboptimal datasets (e.g., with random actions).

Aiming for a provably-efficient approach to utilizing highly suboptimal offline datasets, we use first principles to derive an upper bound on the quality of an imitation learned policy involving three terms corresponding to (1) action representation and (2) action decoder learning on a suboptimal offline dataset, and finally, (3) behavioral cloning (i.e., max-likelihood learning of latent actions) on an expert demonstration dataset. The first term in our bound immediately suggests a practical offline training objective based on a transition dynamics loss using an *factored* transition model. We show that under specific factorizations (e.g., low-dimensional or linear), one can guarantee improved sample efficiency on the expert dataset. Crucially, our mathematical results avoid the potential shortcomings of temporal skill extraction, as our bound is guaranteed to hold even when there is no temporal abstraction in the latent action space.

We translate these mathematical results into an algorithm that we call *Transition-Reparametrized Actions for Imitation Learning* (TRAIL). As shown in Figure 1, TRAIL consists of a pretraining stage (corresponding to the first two terms in our bound) and a downstream imitation learning stage (corresponding to the last term in our bound). During the pretraining stage, TRAIL uses an offline dataset to learn a factored transition model and a paired action decoder. During the downstream imitation learning stage, TRAIL first reparametrizes expert actions into the latent action space according to the learned transition model, and then learns a latent policy via behavioral cloning in the latent action space. During inference, TRAIL uses the imitation learned latent policy and action decoder in conjunction to act in the environment. In practice, TRAIL parametrizes the transition model as an energy-based model (EBM) for flexibility and trains the EBM with a contrastive loss. The EBM enables the low-dimensional factored transition model referenced by our theory, and we also show that one can recover the *linear* transition model in our theory by approximating the EBM with random Fourier features (Rahimi et al., 2007).

To summarize, our contributions include (i) a provably beneficial objective for learning action representations without temporal abstraction and (ii) a practical algorithm for optimizing the proposed objective by learning an EBM or linear transition model. An extensive evaluation on a set of navigation and locomotion tasks demonstrates the effectiveness of the proposed objective. TRAIL's empirical success compared to a variety of existing methods suggests that the benefit of learning *single-step* action representations has been overlooked by previous temporal skill extraction methods. Additionally, TRAIL significantly improves behavioral cloning even when the offline dataset is unimodal or highly suboptimal (e.g., obtained from a random policy), whereas temporal skill extraction methods lead to *degraded* performance in these scenarios. Lastly, we show that TRAIL, without using reward labels, can perform similarly or better than offline reinforcement learning (RL) with orders of magnitude less expert data, suggesting new ways for offline learning of squential decision making policies.

## 2 RELATED WORK

Learning action abstractions is a long standing topic in the hierarchical RL literature (Parr & Russell, 1998; Dieterich et al., 1998; Sutton et al., 1999; Kulkarni et al., 2016; Nachum et al., 2018a). A large body of work focusing on *online skill discovery* have been proposed as a means to improve exploration and sample complexity in online RL. For instance, Eysenbach et al. (2018); Sharma et al. (2019); Gregor et al. (2016); Warde-Farley et al. (2018); Liu et al. (2021) propose to learn a diverse set of skills by maximizing an information theoretic objective. Online skill discovery is also commonly seen in a hierarchical framework that learns a continuous space (Vezhnevets et al., 2017; Hausman et al., 2018; Nachum et al., 2018a; 2019) or a discrete set of lower-level policies (Bacon

et al., 2017; Stolle & Precup, 2002; Peng et al., 2019), upon which higher-level policies are trained to solve specific tasks. Different from these works, we focus on learning action representations *offline* from a fixed suboptimal dataset to accelerate imitation learning.

Aside from online skill discovery, *offline skill extraction* focuses on learning temporally extended action abstractions from a fixed offline dataset. Methods for offline skill extraction generally involve maximum likelihood training of some latent variable models on the offline data, followed by downstream planning (Lynch et al., 2020), imitation learning (Kipf et al., 2019; Ajay et al., 2020; Hakhamaneshi et al., 2021), offline RL (Ajay et al., 2020; Zhou et al., 2020), or online RL (Fox et al., 2017; Krishnan et al., 2017; Shankar & Gupta, 2020; Shankar et al., 2019; Singh et al., 2020; Pertsch et al., 2020; 2021; Wang et al., 2021) in the induced latent action space. Among these works, those that provide a theoretical analysis attribute the benefit of skill extraction predominantly to increased temporal abstraction as opposed to the learned action space being any "easier" to learn from than the raw action space (Ajay et al., 2020; Nachum et al., 2018b). Unlike these methods, our analysis focuses on the advantage of a lower-dimensional reparametrized action space agnostic to temporal abstraction. Our method also applies to offline data that is highly suboptimal (e.g., contains random actions) and potentially unimodal (e.g., without diverse skills to be extracted), which have been considered challenging by previous work (Ajay et al., 2020).

While we focus on reducing the complexity of the action space through the lens of action representation learning, there exists a disjoint set of work that focuses on accelerating RL with *state* representation learning (Singh et al., 1995; Ren & Krogh, 2002; Castro & Precup, 2010; Gelada et al., 2019; Zhang et al., 2020; Arora et al., 2020; Nachum & Yang, 2021), some of which have proposed to extract a latent state space from a learned dynamics model. Analogous to our own derivations, these works attribute the benefit of representation learning to a smaller latent state space reduced from a high-dimensional input state space (e.g., images). Lastly, there exist model-based approaches that utilizes offline data to learn model dynamics which in tern accelerates imitation (Chang et al., 2021; Rafailov et al., 2021). These work differ from our focus of using the offline data to learn latent action space.

## 3 PRELIMINARIES

In this section, we introduce the problem statements for imitation learning and learning-based control, and define relevant notations.

**Markov decision process.**  Consider an MDP (Puterman, 1994) $\mathcal{M} := \langle S, A, \mathcal{R}, \mathcal{T}, \mu, \gamma \rangle$, consisting of a state space $S$, an action space $A$, a reward function $\mathcal{R} : S \times A \to \mathbb{R}$, a transition function $\mathcal{T} : S \times A \to \Delta(S)$[1], an initial state distribution $\mu \in \Delta(S)$, and a discount factor $\gamma \in [0, 1)$ A policy $\pi : S \to \Delta(A)$ interacts with the environment starting at an initial state $s_0 \sim \mu$. An action $a_t \sim \pi(s_t)$ is sampled and applied to the environment at each step $t \geq 0$. The environment produces a scalar reward $\mathcal{R}(s_t, a_t)$ and transitions into the next state $s_{t+1} \sim \mathcal{T}(s_t, a_t)$. Note that we are specifically interested in the imitation learning setting, where the rewards produced by $\mathcal{R}$ are unobserved by the learner. The state visitation distribution $d^\pi(s)$ induced by a policy $\pi$ is defined as $d^\pi(s) := (1 - \gamma) \sum_{t=0}^{\infty} \gamma^t \cdot \Pr[s_t = s | \pi, \mathcal{M}]$. We relax the notation and use $(s, a) \sim d^\pi$ to denote $s \sim d^\pi, a \sim \pi(s)$.

**Learning goal.**  Imitation learning aims to recover an *expert policy* $\pi_*$ with access to only a fixed set of samples from the expert: $\mathcal{D}^{\pi_*} = \{(s_i, a_i)\}_{i=1}^n$ with $s_i \sim d_*^\pi$ and $a_i \sim \pi_*(s_i)$. One approach to imitation learning is to learn a policy $\pi$ that minimizes some discrepancy between $\pi$ and $\pi_*$. In our analysis, we will use the total variation (TV) divergence in state visitation distributions,

$$\text{Diff}(\pi, \pi_*) = D_{\text{TV}}(d^\pi \| d^{\pi_*}),$$

as the way to measure the discrepancy between $\pi$ and $\pi_*$. Our bounds can be easily modified to apply to other divergence measures such as the Kullback–Leibler (KL) divergence or difference in expected future returns. *Behavioral cloning* (BC) (Pomerleau, 1989) solves the imitation learning problem by learning $\pi$ from $\mathcal{D}^{\pi_*}$ via a maximum likelihood objective

$$J_{\text{BC}}(\pi) := \mathbb{E}_{(s,a) \sim (d^{\pi_*}, \pi_*)}[-\log \pi(a|s)],$$

which optimizes an upper bound of $\text{Diff}(\pi, \pi_*)$ defined above (Ross & Bagnell, 2010; Nachum & Yang, 2021):

$$\text{Diff}(\pi, \pi_*) \leq \frac{\gamma}{1 - \gamma} \sqrt{\frac{1}{2} \mathbb{E}_{d^{\pi_*}}[D_{\text{KL}}(\pi_*(s) \| \pi(s))]} = \frac{\gamma}{1 - \gamma} \sqrt{\text{const}(\pi_*) + \frac{1}{2} J_{\text{BC}}(\pi)}.$$

---

[1] $\Delta(\mathcal{X})$ denotes the simplex over a set $\mathcal{X}$.

**BC with suboptimal offline data.** The standard BC objective (i.e., direct max-likelihood on $\mathcal{D}^{\pi_*}$) can struggle to attain good performance when the amount of expert demonstrations is limited (Ross et al., 2011; Tu et al., 2021). We assume access to an additional *suboptimal* offline dataset $\mathcal{D}^{\text{off}} = \{(s_i, a_i, s'_i)\}_{i=1}^m$, where the suboptimality is a result of (i) suboptimal action samples $a_i \sim \text{Unif}_A$ and (ii) lack of reward labels. We use $(s, a, s') \sim d^{\text{off}}$ as a shorthand for simulating finite sampling from $\mathcal{D}^{\text{off}}$ via $s_i \sim d^{\text{off}}, a_i \sim \text{Unif}_A, s'_i \sim \mathcal{T}(s_i, a_i)$, where $d^{\text{off}}$ is an *unknown* offline state distribution. We assume $d^{\text{off}}$ sufficiently covers the expert distribution; i.e., $d^{\pi_*}(s) > 0 \Rightarrow d^{\text{off}}(s) > 0$ for all $s \in S$. The uniform sampling of actions in $\mathcal{D}^{\text{off}}$ is largely for mathematical convenience, and in theory can be replaced with any distribution uniformly bounded from below by $\eta > 0$, and our derived bounds will be scaled by $\frac{1}{|A|\eta}$ as a result. This works focuses on how to utilize such a suboptimal $\mathcal{D}^{\text{off}}$ to provably accelerate BC.

## 4 NEAR-OPTIMAL IMITATION LEARNING WITH REPARAMETRIZED ACTIONS

In this section, we provide a provably-efficient objective for learning action representations from suboptimal data. Our initial derivations (Theorem 1) apply to general policies and latent action spaces, while our subsequent result (Theorem 3) provides improved bounds for specialized settings with continuous latent action spaces. Finally, we present our practical method TRAIL for action representation learning and downstream imitation learning.

### 4.1 PERFORMANCE BOUND WITH REPARAMETRIZED ACTIONS

Despite $\mathcal{D}^{\text{off}}$ being highly suboptimal (e.g., with random actions), the large set of $(s, a, s')$ tuples from $\mathcal{D}^{\text{off}}$ reveals the transition dynamics of the environment, which a latent action space should support. Under this motivation, we propose to learn a *factored* transition model $\overline{\mathcal{T}} := \mathcal{T}_Z \circ \phi$ from the offline dataset $\mathcal{D}^{\text{off}}$, where $\phi : S \times A \to Z$ is an action representaiton function and $\mathcal{T}_Z : S \times Z \to \Delta(S)$ is a latent transition model. Intuitively, good action representations should enable good imitation learning.

We formalize this intuition in the theorem below by establishing a bound on the quality of a learned policy based on (1) an offline pretraining objective for learning $\phi$ and $\mathcal{T}_Z$, (2) an offline decoding objective for learning an action decoder $\pi_\alpha$, and (3) a downstream imitation learning objective for learning a latent policy $\pi_Z$ with respect to latent actions determined by $\phi$.

**Theorem 1.** *Consider an action representation function $\phi : S \times A \to Z$, a factored transition model $\mathcal{T}_Z : S \times Z \to \Delta(S)$, an action decoder $\pi_\alpha : S \times Z \to \Delta(A)$, and a tabular latent policy $\pi_Z : S \to \Delta(Z)$. Define the transition representation error as*

$$J_{\text{T}}(\mathcal{T}_Z, \phi) := \mathbb{E}_{(s,a) \sim d^{\text{off}}} \left[ D_{\text{KL}}(\mathcal{T}(s,a) \| \mathcal{T}_Z(s, \phi(s,a))) \right],$$

*the action decoding error as*

$$J_{\text{DE}}(\pi_\alpha, \phi) := \mathbb{E}_{(s,a) \sim d^{\text{off}}}[- \log \pi_\alpha(a|s, \phi(s,a))],$$

*and the latent behavioral cloning error as*

$$J_{\text{BC},\phi}(\pi_Z) := \mathbb{E}_{(s,a) \sim (d^{\pi_*}, \pi_*)}[- \log \pi_Z(\phi(s,a)|s)].$$

*Then the TV divergence between the state visitation distributions of $\pi_\alpha \circ \pi_Z : S \to \Delta(A)$ and $\pi_*$ can be bounded as*

$$\text{Diff}(\pi_\alpha \circ \pi_Z, \pi_*) \leq$$

*Pretraining*
$$C_1 \cdot \sqrt{\frac{1}{2} \underbrace{\mathbb{E}_{(s,a) \sim d^{\text{off}}} \left[ D_{\text{KL}}(\mathcal{T}(s,a) \| \mathcal{T}_Z(s, \phi(s,a))) \right]}_{= J_{\text{T}}(\mathcal{T}_Z, \phi)}} \quad (1)$$

$$+ C_2 \cdot \sqrt{\frac{1}{2} \underbrace{\mathbb{E}_{s \sim d^{\text{off}}} [\max_{z \in Z} D_{\text{KL}}(\pi_{\alpha^*}(s,z) \| \pi_\alpha(s,z))]}_{\approx \text{const}(d^{\text{off}}, \phi) + J_{\text{DE}}(\pi_\alpha, \phi)}} \quad (2)$$

*Downstream Imitation*
$$+ C_3 \cdot \sqrt{\frac{1}{2} \underbrace{\mathbb{E}_{s \sim d^{\pi_*}} [D_{\text{KL}}(\pi_{*,Z}(s) \| \pi_Z(s))]}_{= \text{const}(\pi_*, \phi) + J_{\text{BC},\phi}(\pi_Z)}}, \quad (3)$$

*where $C_1 = \gamma |A| (1-\gamma)^{-1} (1 + D_{\chi^2}(d^{\pi_*} \| d^{\text{off}})^{\frac{1}{2}})$, $C_2 = \gamma (1-\gamma)^{-1} (1 + D_{\chi^2}(d^{\pi_*} \| d^{\text{off}})^{\frac{1}{2}})$, $C_3 = \gamma (1-\gamma)^{-1}$, $\pi_{\alpha^*}$ is the optimal action decoder for a specific data distribution $d^{\text{off}}$ and a*

*specific $\phi$:*

$$\pi_{\alpha^*}(a|s,z) = \frac{d^{\mathrm{off}}(s,a) \cdot \mathbb{1}[z = \phi(s,a)]}{\sum_{a' \in A} d^{\mathrm{off}}(s,a') \cdot \mathbb{1}[z = \phi(s,a')]},$$

*and $\pi_{*,Z}$ is the marginalization of $\pi_*$ onto $Z$ according to $\phi$:*

$$\pi_{*,Z}(z|s) := \sum_{a \in A, z = \phi(s,a)} \pi_*(a|s).$$

Theorem 1 essentially decomposes the imitation learning error into (1) a transition-based representation error $J_{\mathrm{T}}$, (2) an action decoding error $J_{\mathrm{DE}}$, and (3) a latent behavioral cloning error $J_{\mathrm{BC},\phi}$. Notice that only (3) requires expert data $\mathcal{D}^{\pi_*}$; (1) and (2) are trained on the large offline data $\mathcal{D}^{\mathrm{off}}$. By choosing $|Z|$ that is smaller than $|A|$, fewer demonstrations are needed to achieve small error in $J_{\mathrm{BC},\phi}$ compared to vanilla BC with $J_{\mathrm{BC}}$. The Pearson $\chi^2$ divergence term $D_{\chi^2}(d^{\pi_*}\|d^{\mathrm{off}})$ in $C_1$ and $C_2$ accounts for the difference in state visitation between the expert and offline data. In the case where $d^{\pi_*}$ differs too much from $d^{\mathrm{off}}$, known as the distribution shift problem in offline RL (Levine et al., 2020), the errors from $J_{\mathrm{T}}$ and $J_{\mathrm{DE}}$ are amplified and the terms (1) and (2) in Theorem 1 dominate. Otherwise, as $J_{\mathrm{T}} \to 0$ and $\pi_\alpha, \phi \to \arg\min J_{\mathrm{DE}}$, optimizing $\pi_Z$ in the latent action space is guaranteed to optimize $\pi$ in the original action space.

**Sample Complexity**    To formalize the intuition that a smaller latent action space $|Z| < |A|$ leads to more sample efficient downstream behavioral cloning, we provide the following theorem in the tabular action setting. First, assume access to an oracle latent action representation function $\phi_{orcl} := \mathcal{OPT}_\phi(\mathcal{D}^{\mathrm{off}})$ which yields pretraining errors (1)($\phi_{orcl}$) and (2)($\phi_{orcl}$) in Theorem 1. For downstream behavioral cloning, we consider learning a tabular $\pi_Z$ on $\mathcal{D}^{\pi_*}$ with $n$ expert samples. We can bound the expected difference between a latent policy $\pi_{\phi_{orcl},Z}$ with respect to $\phi_{orcl}$ and $\pi_*$ as follows.

**Theorem 2.** *Let $\phi_{orcl} := \mathcal{OPT}_\phi(\mathcal{D}^{\mathrm{off}})$ and $\pi_{orcl,Z}$ be the latent BC policy with respect to $\phi_{orcl}$. We have,*

$$\mathbb{E}_{\mathcal{D}^{\pi_*}}[\mathrm{Diff}(\pi_{\phi_{orcl},Z}, \pi_*)] \le (1)(\phi_{orcl}) + (2)(\phi_{orcl}) + C_3 \cdot \sqrt{\frac{|Z||S|}{n}},$$

*where $C_3$ is the same as in Theorem 1.*

We can contrast this bound to its form in the vanilla BC setting, for which $|Z| = |A|$ and both (1)($\phi_{orcl}$) and (2)($\phi_{orcl}$) are zero. We can expect an improvement in sample complexity from reparametrized actions when the errors in (1) and (2) are small and $|Z| < |A|$.

## 4.2 Linear Transition Models with Deterministic Latent Policy

Theorem 1 has introduced the notion of a latent expert policy $\pi_{*,Z}$, and minimizes the KL divergence between $\pi_{*,Z}$ and a *tabular* latent policy $\pi_Z$. However, it is not immediately clear, in the case of continuous latent actions, how to ensure that the latent policy $\pi_Z$ is expressive enough to capture any $\pi_{*,Z}$. In this section, we provide guarantees for recovering stochastic expert policies with continuous latent action space under a linear transition model.

Consider a *continuous* latent space $Z \subset \mathbb{R}^d$ and a *deterministic* latent policy $\pi_\theta(s) = \theta_s$ for some $\theta \in \mathbb{R}^{d \times |S|}$. While a deterministic $\theta$ in general cannot capture a stochastic $\pi_*$, we show that under a linear transition model $\mathcal{T}_Z(s'|s, \phi(s,a)) = w(s')^\top \phi(s,a)$, there always exists a deterministic policy $\pi_\theta : S \to \mathbb{R}^d$, such that $\theta_s = \pi_{*,Z}(s), \forall s \in S$. This means that our scheme for offline pretraining paired with downstream imitation learning can *provably* recover any expert policy $\pi_*$ from a deterministic $\pi_\theta$, regardless of whether $\pi_*$ is stochastic.

**Theorem 3.** *Let $\phi : S \times A \to Z$ for some $Z \subset \mathbb{R}^d$ and suppose there exist $w : S \to \mathbb{R}^d$ such that $\mathcal{T}_Z(s'|s, \phi(s,a)) = w(s')^\top \phi(s,a)$ for all $s, s' \in S, a \in A$. Let $\pi_\alpha : S \times Z \to \Delta(A)$ be an action decoder, $\pi : S \to \Delta(A)$ be any policy in $\mathcal{M}$ and $\pi_\theta : S \to \mathbb{R}^d$ be a deterministic latent policy for some $\theta \in \mathbb{R}^{d \times |S|}$. Then,*

$$\mathrm{Diff}(\pi_\alpha \circ \pi_\theta, \pi_*) \le (1)(\mathcal{T}_Z, \phi) + (2)(\pi_\alpha, \phi)$$

$$\begin{array}{c} \textit{Downstream} \\ \textit{Imitation} \end{array} \left\{ + C_4 \cdot \left\| \frac{\partial}{\partial \theta} \mathbb{E}_{s \sim d^{\pi_*}, a \sim \pi_*(s)}[(\theta_s - \phi(s,a))^2] \right\|_1 \right., \tag{4}$$

*where $C_4 = \frac{1}{4}|S|\|w\|_\infty$, (1) and (2) corresponds to the first and second terms in the bound in Theorem 1.*

By replacing term (3) in Theorem 1 that corresponds to behavioral cloning in the latent action space by term (4) in Theorem 3 that is a convex function unbounded in all directions, we are guaranteed that $\pi_\theta$ is provably optimal regardless of the form of $\pi_*$ and $\pi_{*,Z}$. Note that the downstream imitation learning objective implied by term (4) is simply the mean squared error between actions $\theta_s$ chosen by $\pi_\theta$ and reparameterized actions $\phi(s, a)$ appearing in the expert dataset.

## 4.3 TRAIL: Reparametrized Actions and Imitation Learning in Practice

In this section, we describe our learning framework, Transition-Reparametrized Actions for Imitation Learning (TRAIL). TRAIL consists of two training stages: pretraining and downstream behavioral cloning. During pretraining, TRAIL learns $\mathcal{T}_Z$ and $\phi$ by minimizing $J_\text{T}(\mathcal{T}_Z, \phi) = \mathbb{E}_{(s,a)\sim d^\text{off}}[D_\text{KL}(\mathcal{T}(s,a)\|\mathcal{T}_Z(s,\phi(s,a)))]$. Also during pretraining, TRAIL learns $\pi_\alpha$ and $\phi$ by minimizing $J_\text{DE}(\pi_\alpha, \phi) := \mathbb{E}_{(s,a)\sim d^\text{off}}[-\log\pi_\alpha(a|s,\phi(s,a))]$. TRAIL parametrizes $\pi_\alpha$ as a multivariate Gaussian distribution. Depending on whether $\mathcal{T}_Z$ is defined according to Theorem 1 or Theorem 3, we have either TRAIL EBM or TRAIL linear.

**TRAIL EBM for Theorem 1.** In the tabular action setting that corresponds to Theorem 1, to ensure that the factored transition model $\mathcal{T}_Z$ is flexible to capture any complex (e.g., multi-modal) transitions in the offline dataset, we propose to use an energy-based model (EBM) to parametrize $\mathcal{T}_Z(s'|s,\phi(s,a))$,

$$\mathcal{T}_Z(s'|s,\phi(s,a)) \propto \rho(s')\exp(-\|\phi(s,a) - \psi(s')\|^2), \tag{5}$$

where $\rho$ is a fixed distribution over $S$ and $\psi : S \to Z$ is a function of $s'$. In our implementation we set $\rho$ to be the distribution of $s'$ in $d^\text{off}$, which enables a practical learning objective for $\mathcal{T}_Z$ by minimizing $\mathbb{E}_{(s,a)\sim d^\text{off}}[D_\text{KL}(\mathcal{T}(s,a)\|\mathcal{T}_Z(s,\phi(s,a)))]$ in Theorem 1 using a contrastive loss:

$$\mathbb{E}_{d^\text{off}}[-\log\mathcal{T}_Z(s'|s,\phi(s,a)))] = \text{const}(d^\text{off}) + \frac{1}{2}\mathbb{E}_{d^\text{off}}[\|\phi(s,a) - \psi(s')\|^2]$$
$$+ \log\mathbb{E}_{\tilde{s}'\sim\rho}[\exp\{-\frac{1}{2}\|\phi(s,a) - \psi(\tilde{s}')\|^2\}].$$

During downstream behavioral cloning, TRAIL EBM learns a latent Gaussian policy $\pi_Z$ by minimizing $J_{\text{BC},\phi}(\pi_Z) = \mathbb{E}_{(s,a)\sim(d^{\pi_*},\pi_*)}[-\log\pi_Z(\phi(s,a)|s)]$ with $\phi$ fixed. During inference, TRAIL EBM first samples a latent action according to $z \sim \pi_Z(s)$, and decodes the latent action using $a \sim \pi_\alpha(s,z)$ to act in an environment. Figure 1 describes this process pictorially.

**TRAIL Linear for Theorem 3.** In the continuous latent action setting that corresponds to Theorem 3, we propose TRAIL linear, an approximation of TRAIL EBM, to enable learning *linear* transition models required by Theorem 3. Specifically, we first learn $f, g$ that parameterize an energy-based transition model $\overline{\mathcal{T}}(s'|s,a) \propto \rho(s')\exp\{-\|f(s,a) - g(s')\|^2/2\}$ using the same contrastive loss as above (replacing $\phi$ and $\psi$ by $f$ and $g$), and then apply random Fourier features (Rahimi et al., 2007) to recover $\bar{\phi}(s,a) = \cos(Wf(s,a) + b)$, where $W$ is a $d \times k$ matrix with entries sampled from a unit Gaussian and $b$ a vector with entries sampled uniformly from $[0, 2\pi]$. $W$ and $b$ are implemented as an untrainable neural network layer on top of $f$. This results in an approximate linear transition model,

$$\overline{\mathcal{T}}(s'|s,a) \propto \rho(s')\exp\{-\|f(s,a) - g(s')\|^2/2\} \propto \bar{\psi}(s')^\top\bar{\phi}(s,a).$$

During downstream behavioral cloning, TRAIL linear learns a deterministic policy $\pi_\theta$ in the continuous latent action space determined by $\bar{\phi}$ via minimizing $\left\|\frac{\partial}{\partial\theta}\mathbb{E}_{s\sim d^{\pi_*},a\sim\pi_*(s)}[(\theta_s - \bar{\phi}(s,a))^2]\right\|_1$ with $\bar{\phi}$ fixed. During inference, TRAIL linear first determines the latent action according to $z = \pi_\theta(s)$, and decodes the latent action using $a \sim \pi_\alpha(s,z)$ to act in an environment.

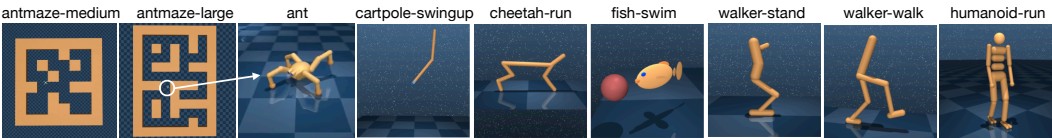

Figure 2: Tasks for our empirical evaluation. We include the challenging AntMaze navigation tasks from D4RL (Fu et al., 2020) and low (1-DoF) to high (21-DoF) dimensional locomotaion tasks from DeepMind Control Suite (Tassa et al., 2018).

Figure 3: Average success rate ($\%$) over 4 seeds of TRAIL EBM (Theorem 1) and temporal skill extraction methods – SkiLD (Pertsch et al., 2021), SPiRL (Pertsch et al., 2020), and OPAL (Ajay et al., 2020) – pretrained on suboptimal $\mathcal{D}^{\text{off}}$. Baseline BC corresponds to direct behavioral cloning of expert $\mathcal{D}^{\pi_*}$ without latent actions.

## 5 EXPERIMENTAL EVALUATION

We now evaluate TRAIL on a set of navigation and locomotion tasks (Figure 2). Our evaluation is designed to study how well TRAIL can improve imitation learning with limited expert data by leveraging available suboptimal offline data. We evaluate the improvement attained by TRAIL over vanilla BC, and additionally compare TRAIL to previously proposed temporal skill extraction methods. Since there is no existing benchmark for imitation learning with suboptimal offline data, we adapt existing datasets for offline RL, which contain suboptimal data, and augment them with a small amount of expert data for downstream imitation learning.

### 5.1 EVALUATING NAVIGATION WITHOUT TEMPORAL ABSTRACTION

**Description and Baselines.** We start our evaluation on the AntMaze task from D4RL (Fu et al., 2020), which has been used as a testbed by recent works on temporal skill extraction for few-shot imitation (Ajay et al., 2020) and RL (Ajay et al., 2020; Pertsch et al., 2020; 2021). We compare TRAIL to OPAL (Ajay et al., 2020), SkiLD (Pertsch et al., 2021), and SPiRL (Pertsch et al., 2020), all of which use an offline dataset to extract temporally extended (length $t = 10$) skills to form a latent action space for downstream learning. SkiLD and SPiRL are originally designed only for downstream RL, so we modify them to support downstream imitation learning as described in Appendix C. While a number of other works have also proposed to learn primitives for hierarchical imitation (Kipf et al., 2019; Hakhamaneshi et al., 2021) and RL (Fox et al., 2017; Krishnan et al., 2017; Shankar et al., 2019; Shankar & Gupta, 2020; Singh et al., 2020), we chose OPAL, SkiLD, and SPiRL for comparison because they are the most recent works in this area with reported results that suggest these methods are state-of-the-art, especially in learning from *suboptimal* offline data based on D4RL. To construct the suboptimal and expert datasets, we follow the protocol in Ajay et al. (2020), which uses the full `diverse` or `play` D4RL AntMaze datasets as the suboptimal offline data, while using a set of $n = 10$ expert trajectories (navigating from one corner of the maze to the opposite corner) as the expert data. The `diverse` and `play` datasets are suboptimal in the corner-to-corner navigation task, as they only contain data that navigates to random or fixed locations different from task evaluation.

**Implementation Details.** For TRAIL, we parameterize $\phi(s, a)$ and $\psi(s')$ using separate feed-forward neural networks (see details in Appendix C) and train the transition EBM via the contrastive objective described in Section 4.3. We parametrize both the action decoder $\pi_\alpha$ and the latent $\pi_Z$ using multivariate Gaussian distributions with neural-network approximated mean and variance. For the temporal skill extraction methods, we implement the trajectory encoder using a bidirectional RNN and parametrize skill prior, latent policy, and action decoder as Gaussians following Ajay et al. (2020). We adapt SPiRL and SkiLD for imitation learning by including the KL Divergence term between the latent policy and the skill prior during downstream behavioral cloning (see details in Appendix C). We do a search on the extend of temporal abstraction, and found $t = 10$ to work the best as reported in these papers' maze experiments. We also experimented with a version of vanilla BC pretrained on the suboptimal data and fine-tuned on expert data for fair comparison, which did not show a significant difference from directly training vanilla BC on expert data.

**Results.** Figure 3 shows the average performance of TRAIL in terms of task success rate (out of 100%) compared to the prior methods. Since all of the prior methods are proposed in terms of temporal abstraction, we evaluate them both with the default temporal abstract, $t = 10$, as well as without temporal abstraction, corresponding to $t = 1$. Note that TRAIL uses *no* temporal abstraction. We

Figure 4: Average rewards (over $4$ seeds) of TRAIL EBM (Theorem 1), TRAIL linear (Theorem 3), and baseline methods when using a variety of unimodal (`ant-medium`), low-quality (`ant-medium-replay`), and random (`ant-random`) offline datasets $\mathcal{D}^{\text{off}}$ paired with a smaller expert dataset $\mathcal{D}^{\pi_*}$ (either 10k or 25k expert transitions).

find that on the simpler `antmaze-medium` task, TRAIL trained on a single-step transition model performs similarly to the set of temporal skill extraction methods with $t = 10$. However, these skill extraction methods experience a degradation in performance when temporal abstraction is removed ($t = 1$). This corroborates the existing theory in these works (Ajay et al., 2020), which attributes their benefits predominantly to temporal abstraction rather than producing a latent action space that is "easier" to learn. Meanwhile, TRAIL is able to excel without any temporal abstraction.

These differences become even more pronounced on the harder `antmaze-large` tasks. We see that TRAIL maintains significant improvements over vanilla BC, whereas temporal skill extraction fails to achieve good performance even with $t = 10$. These results suggest that TRAIL attains significant improvement specifically from utilizing the suboptimal data for learning suitable action representations, rather than simply from providing temporal abstraction. Of course, this does not mean that temporal abstraction is never helpful. Rather, our results serve as evidence that suboptimal data can be useful for imitation learning not just by providing temporally extended skills, but by actually reformulating the action space to make imitation learning easier and more efficient.

## 5.2 EVALUATING LOCOMOTION WITH HIGHLY SUBOPTIMAL OFFLINE DATA

**Description.** The performance of TRAIL trained on a *single-step* transition model in the previous section suggests that learning single-step latent action representations can benefit a broader set of tasks for which temporal abstraction may not be helpful, e.g., when the offline data is highly suboptimal (with near-random actions) or unimodal (collected by a single stationary policy). In this section, we consider a Gym-MuJoCo task from D4RL using the same 8-DoF quadruped ant robot as the previously evaluated navigation task. We first learn action representations from the `medium`, `medium-replay`, or `random` datasets, and imitate from $1\%$ or $2.5\%$ of the `expert` datasets from D4RL. The `medium` dataset represents data collected from a mediocre stationary policy (exhibiting unimodal behavior), and the `random` dataset is collected by a randomly initialized policy and is hence highly suboptimal.

**Implementation Details.** For this task, we additionally train a linear version of TRAIL by approximating the transition EBM using random Fourier features (Rahimi et al., 2007) and learn a *deterministic* latent policy following Theorem 3. Specifically, we use separate feed-forward networks to parameterize $f(s, a)$ and $g(s')$, and extract action representations using $\phi(s, a) = \cos(Wf(s, a) + b)$, where $W, b$ are untrainable randomly initialized variables as described in Section 4.3. Different from TRAIL EBM which parametrizes $\pi_Z$ as a Gaussian, TRAIL linear parametrizes the *deterministic* $\pi_\theta$ using a feed-forward neural network.

**Results.** Our results are shown in Figure 4. Both the EBM and linear versions of TRAIL consistently improve over baseline BC, whereas temporal skill extraction methods generally lead to worse performance regardless of the extent of abstraction, likely due to the degenerate effect (i.e., latent skills being ignored by a flexible action decoder) resulted from unimodal offline datasets as discussed in (Ajay et al., 2020). Surprisingly, TRAIL achieves a significant performance boost even when latent actions are learned from the `random` dataset, suggesting the benefit of learning action representations from transition models when the offline data is highly suboptimal. Additionally, the linear variant of TRAIL performs slightly better than the EBM variant when the expert sample size is small (i.e., 10k), suggesting the benefit of learning deterministic latent policies from Theorem 3 when the environment is effectively approximated by a linear transition model.

Figure 5: Average task rewards (over $4$ seeds) of TRAIL EBM (Theorem 1), TRAIL linear (Theorem 3), and OPAL (other temporal methods are included in Appendix D) pretrained on the bottom $80\%$ of the RL Unplugged datasets followed by behavioral cloning in the latent action space on $\frac{1}{10}$ of the top $20\%$ of the RL Unplugged datasets following the setup in Zolna et al. (2020). Baseline BC achieves low rewards due to the small expert sample size. Dotted lines denote the performance of CRR (Wang et al., 2020), an offline RL method trained on the full RL Unplugged datasets with reward labels.

## 5.3 EVALUATION ON DEEPMIND CONTROL SUITE

**Description.** Having witnessed the improvement TRAIL brings to behavioral cloning on AntMaze and MuJoCo Ant, we wonder how TRAIL perform on a wider spectrum of locomotion tasks with various degrees of freedom. We consider $6$ locomotion tasks from the DeepMind Control Suite (Tassa et al., 2018) ranging from simple (e.g., 1-DoF `cartople-swingup`) to complex (e.g., 21-DoF `humanoid-run`) tasks. Following the setup in Zolna et al. (2020), we take $\frac{1}{10}$ of the trajectories whose episodic reward is among the top $20\%$ of the open source RL Unplugged datasets (Gulcehre et al., 2020) as expert demonstrations (see numbers of expert trajectories in Appendix C), and the bottom $80\%$ of RL Unplugged as the suboptimal offline data. For completeness, we additionally include comparison to Critic Regularized Regression (CRR) (Wang et al., 2020), an offline RL method with competitive performance on these tasks. CRR is trained on the full RL Unplugged datasets (i.e., combined suboptimal and expert datasets) with reward labels.

**Results.** Figure 5 shows the comparison results. TRAIL outperforms temporal extraction methods on both low-dimensional (e.g., `cartpole-swingup`) and high-dimensional (`humanoid-run`) tasks. Additionally, TRAIL performs similarly to or better than CRR on $4$ out of the $6$ tasks despite not using any reward labels, and only slightly worse on `humanoid-run` and `walker-walk`. To test the robustness of TRAIL when the offline data is highly suboptimal, we further reduce the size and quality of the offline data to the bottom $5\%$ of the original RL Unplugged datasets. As shown in Figure 6 in Appndix D, the performance of temporal skill extraction declines in `fish-swim`, `walker-stand`, and `walker-walk` due to this change in offline data quality, whereas TRAIL maintains the same performance as when the bottom $80\%$ data was used, suggesting that TRAIL is more robust to low-quality offline data.

This set of results suggests a promising direction for offline learning of sequential decision making policies, namely to learn latent actions from abundant low-quality data and behavioral cloning in the latent action space on scarce high-quality data. Notably, compared to offline RL, this approach is applicable to settings where data quality cannot be easily expressed through a scalar reward.

## 6 CONCLUSION

We have derived a near-optimal objective for learning a latent action space from suboptimal offline data that provably accelerates downstream imitation learning. To learn this objective in practice, we propose transition-reparametrized actions for imitation learning (TRAIL), a two-stage framework that first pretrains a factored transition model from offline data, and then uses the transition model to reparametrize the action space prior to behavioral cloning. Our empirical results suggest that TRAIL can improve imitation learning drastically, even when pretrained on highly suboptimal data (e.g., data from a random policy), providing a new approach to imitation learning through a combination of pretraining on task-agnostic or suboptimal data and behavioral cloning on limited expert datasets. That said, our approach to action representation learning is not necessarily specific to imitation learning, and insofar as the reparameterized action space simplifies downstream control problems, it could also be combined with reinforcement learning in future work. More broadly, studying how learned action reparameterization can accelerate various facets of learning-based control represents an exciting future direction, and we hope that our results provide initial evidence of such a potential.

ACKNOWLEDGMENTS

We thank Dale Schuurmans and Bo Dai for valuable discussions. We thank Justin Fu, Anurag Ajay, and Konrad Zolna for assistance in setting up evaluation tasks.

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

# Appendix

## A  PROOFS FOR FOUNDATIONAL LEMMAS

**Lemma 4.** *If $\pi_1$ and $\pi_2$ are two policies in $\mathcal{M}$ and $d^{\pi_1}(s)$ and $d^{\pi_2}(s)$ are the state visitation distributions induced by policy $\pi_1$ and $\pi_2$ where $d^{\pi}(s) := (1 - \gamma) \sum_{t=0} \gamma^t \cdot \Pr[s_t = s | \pi, \mathcal{M}]$. Define $\mathrm{Diff}(\pi_2, \pi_1) = D_{\mathrm{TV}}(d^{\pi_2} \| d^{\pi_1})$ then*

$$\mathrm{Diff}(\pi_2, \pi_1) \leq \frac{\gamma}{1 - \gamma} \mathrm{Err}_{d^{\pi_1}}(\pi_1, \pi_2, \mathcal{T}), \tag{6}$$

*where*

$$\mathrm{Err}_{d^{\pi_1}}(\pi_1, \pi_2, \mathcal{T}) := \frac{1}{2} \sum_{s' \in S} \left| \mathbb{E}_{s \sim d^{\pi_1}, a_1 \sim \pi_1(s), a_2 \sim \pi_2(s)}[\mathcal{T}(s'|s, a_1) - \mathcal{T}(s'|s, a_2)] \right|. \tag{7}$$

*is the TV-divergence between $\mathcal{T} \circ \pi_1 \circ d^{\pi_1}$ and $\mathcal{T} \circ \pi_2 \circ d^{\pi_1}$.*

*Proof.* Following similar derivations in Achiam et al. (2017); Nachum et al. (2018b), we express $D_{\mathrm{TV}}(d^{\pi_2} \| d^{\pi_1})$ in linear operator notation:

$$\mathrm{Diff}(\pi_2, \pi_1) = D_{\mathrm{TV}}(d^{\pi_2} \| d^{\pi_1}) = \frac{1}{2} \mathbf{1} |(1 - \gamma)(I - \gamma \mathcal{T} \Pi_2)^{-1} \mu - (1 - \gamma)(I - \gamma \mathcal{T} \Pi_1)^{-1} \mu|, \tag{8}$$

where $\Pi_1, \Pi_2$ are linear operators $S \to S \times A$ such that $\Pi_i \nu(s, a) = \pi_i(a|s) \nu(s)$ and $\mathbf{1}$ is an all ones row vector of size $|S|$. Notice that $d^{\pi_1}$ may be expressed in this notation as $(1 - \gamma)(I - \gamma \mathcal{T} \Pi_1)^{-1} \mu$. We may re-write the above term as

$$\frac{1}{2} \mathbf{1} |(1 - \gamma)(I - \gamma \mathcal{T} \Pi_2)^{-1}((I - \gamma \mathcal{T} \Pi_1) - (I - \gamma \mathcal{T} \Pi_2))(I - \gamma \mathcal{T} \Pi_1)^{-1} \mu|$$

$$= \gamma \cdot \frac{1}{2} \mathbf{1} |(I - \gamma \mathcal{T} \Pi_2)^{-1} (\mathcal{T} \Pi_2 - \mathcal{T} \Pi_1) d^{\pi_1}|. \tag{9}$$

Using matrix norm inequalities, we bound the above by

$$\gamma \cdot \frac{1}{2} \|(I - \gamma \mathcal{T} \Pi_2)^{-1}\|_{1,\infty} \cdot \mathbf{1} |(\mathcal{T} \Pi_2 - \mathcal{T} \Pi_1) d^{\pi_1}|. \tag{10}$$

Since $\mathcal{T} \Pi_2$ is a stochastic matrix, $\|(I - \gamma \mathcal{T} \Pi_2)^{-1}\|_{1,\infty} \leq \sum_{t=0}^{\infty} \gamma^t \|\mathcal{T} \Pi_2\|_{1,\infty} = (1 - \gamma)^{-1}$. Thus, we bound the above by

$$\frac{\gamma}{2(1 - \gamma)} \mathbf{1} |(\mathcal{T} \Pi_2 - \mathcal{T} \Pi_1) d^{\pi_1}| = \frac{\gamma}{1 - \gamma} \mathrm{Err}_{d^{\pi_1}}(\pi_1, \pi_2, \mathcal{T}), \tag{11}$$

and so we immediately achieve the desired bound in equation 6. $\square$

The divergence bound above relies on the true transition model $\mathcal{T}$ which is not available to us. We now introduce an approximate transition model $\overline{\mathcal{T}}$ to proxy $\mathrm{Err}_{d^{\pi_1}}(\pi_1, \pi_2, \mathcal{T})$.

**Lemma 5.** *For $\pi_1$ and $\pi_2$ two policies in $\mathcal{M}$ and any transition model $\overline{\mathcal{T}}(\cdot|s, a)$ we have,*

$$\mathrm{Err}_{d^{\pi_1}}(\pi_1, \pi_2, \mathcal{T}) \leq |A| \mathbb{E}_{(s,a) \sim (d^{\pi_1}, \mathrm{Unif}_A)}[D_{\mathrm{TV}}(\mathcal{T}(s, a) \| \overline{\mathcal{T}}(s, a))] + \mathrm{Err}_{d^{\pi_1}}(\pi_1, \pi_2, \overline{\mathcal{T}}). \tag{12}$$

*Proof.*

$$\mathrm{Err}_{d^{\pi_1}}(\pi_1, \pi_2, \mathcal{T}) = \frac{1}{2} \sum_{s' \in S} \left| \mathbb{E}_{s \sim d^{\pi_1}, a_1 \sim \pi_1(s), a_2 \sim \pi_2(s)}[\mathcal{T}(s'|s, a_1) - \mathcal{T}(s'|s, a_2)] \right| \tag{13}$$

$$= \frac{1}{2} \sum_{s' \in S} \left| \sum_{a \in A} \mathbb{E}_{s \sim d^{\pi_1}}[\mathcal{T}(s'|s, a) \pi_1(a|s) - \mathcal{T}(s'|s, a) \pi_2(a|s)] \right| \tag{14}$$

$$= \frac{1}{2} \sum_{s' \in S} \left| \sum_{a \in A} \mathbb{E}_{s \sim d^{\pi_1}}[(\mathcal{T}(s'|s, a) - \overline{\mathcal{T}}(s'|s, a))(\pi_1(a|s) - \pi_2(a|s)) + \overline{\mathcal{T}}(s'|s, a)(\pi_1(a|s) - \pi_2(a|s))] \right| \tag{15}$$

$$\leq \frac{1}{2} \sum_{s' \in S} \left| \sum_{a \in A} \mathbb{E}_{s \sim d^{\pi_1}}[(\mathcal{T}(s'|s, a) - \overline{\mathcal{T}}(s'|s, a))(\pi_1(a|s) - \pi_2(a|s))] \right| + \mathrm{Err}_{d^{\pi_1}}(\pi_1, \pi_2, \overline{\mathcal{T}}) \tag{16}$$

$$\leq \frac{1}{2} \sum_{s' \in S} \sum_{a \in A} \mathbb{E}_{s \sim d^{\pi_1}}[|(\mathcal{T}(s'|s, a) - \overline{\mathcal{T}}(s'|s, a))(\pi_1(a|s) - \pi_2(a|s))|] + \mathrm{Err}_{d^{\pi_1}}(\pi_1, \pi_2, \overline{\mathcal{T}}) \tag{17}$$

$$\leq |A| \mathbb{E}_{(s,a) \sim (d^{\pi_1}, \mathrm{Unif}_A)}[D_{\mathrm{TV}}(\mathcal{T}(s'|s, a) \| \overline{\mathcal{T}}(s'|s, a)]] + \mathrm{Err}_{d^{\pi_1}}(\pi_1, \pi_2, \overline{\mathcal{T}}), \tag{18}$$

and we arrive at the inequality as desired where the last step comes from $D_{\mathrm{TV}}(\mathcal{T}(s,a)\|\overline{\mathcal{T}}(s,a)) = \frac{1}{2}\sum_{s'\in S}|\mathcal{T}(s'|s,a) - \overline{\mathcal{T}}(s'|s,a)|$. $\qquad\square$

Now we introduce a representation function $\phi : S \times A \to Z$ and show how the error above may be reduced when $\overline{\mathcal{T}}(s,a) = \mathcal{T}_Z(s,\phi(s,a))$:

**Lemma 6.** *Let $\phi : S \times A \to Z$ for some space $Z$ and suppose there exists $\mathcal{T}_Z : S \times Z \to \Delta(S)$ such that $\overline{\mathcal{T}}(s,a) = \mathcal{T}_Z(s,\phi(s,a))$ for all $s \in S, a \in A$. Then for any policies $\pi_1, \pi_2$,*

$$\mathrm{Err}_{d^{\pi_1}}(\pi_1, \pi_2, \overline{\mathcal{T}})] \leq \mathbb{E}_{s\sim d^{\pi_1}}[D_{\mathrm{TV}}(\pi_{1,Z}\|\pi_{2,Z})], \tag{19}$$

*where $\pi_{k,Z}(z|s)$ is the marginalization of $\pi_k$ onto $Z$:*

$$\pi_{k,Z}(z|s) := \sum_{a\in A, z=\phi(s,a)} \pi_k(a|s) \tag{20}$$

*for all $z \in Z, k \in \{1,2\}$.*

*Proof.*

$$\frac{1}{2}\sum_{s'\in S}\left|\mathbb{E}_{s\sim d^{\pi_1},a_1\sim\pi_1(s),a_2\sim\pi_2(s)}[\overline{\mathcal{T}}(s'|s,a_1) - \overline{\mathcal{T}}(s'|s,a_2)]\right| \tag{21}$$

$$= \frac{1}{2}\sum_{s'\in S}\left|\sum_{s\in S, a\in A}\mathcal{T}_Z(s'|s,\phi(s,a))\pi_1(a|s)d^{\pi_1}(s) - \sum_{s\in S, a\in A}\mathcal{T}_Z(s'|s,\phi(s,a))\pi_2(a|s)d^{\pi_1}(s)\right|$$

$$= \frac{1}{2}\sum_{s'\in S}\left|\sum_{s\in S, z\in Z}\mathcal{T}_Z(s'|s,z)\sum_{\substack{a\in A,\\ \phi(s,a)=z}}\pi_1(a|s)d^{\pi_1}(s) - \sum_{s\in S, z\in Z}\mathcal{T}_Z(s'|s,z)\sum_{\substack{a\in A,\\ \phi(s,a)=z}}\pi_2(a|s)d^{\pi_1}(s)\right|$$

$$= \frac{1}{2}\sum_{s'\in S}\left|\sum_{s\in S, z\in Z}\mathcal{T}_Z(s'|s,z)\pi_{1,Z}(z|s)d^{\pi_1}(s) - \sum_{s\in S, z\in Z}\mathcal{T}_Z(s'|s,z)\pi_{2,Z}(z|s)d^{\pi_1}(s)\right|$$

$$= \frac{1}{2}\sum_{s'\in S}\left|\mathbb{E}_{s\sim d^{\pi_1}}\left[\sum_{z\in Z}\mathcal{T}_Z(s'|s,z)(\pi_{1,Z}(z|s) - \pi_{2,Z}(z|s))\right]\right| \tag{22}$$

$$\leq \frac{1}{2}\mathbb{E}_{s\sim d^{\pi_1}}\left[\sum_{z\in Z}\sum_{s'\in S}\mathcal{T}_Z(s'|s,z)|\pi_{1,Z}(z|s) - \pi_{2,Z}(z|s)|\right] \tag{23}$$

$$= \frac{1}{2}\mathbb{E}_{s\sim d^{\pi_1}}\left[\sum_{z\in Z}|\pi_{1,Z}(z|s) - \pi_{2,Z}(z|s)|\right] \tag{24}$$

$$= \mathbb{E}_{s\sim d^{\pi_1}}[D_{\mathrm{TV}}(\pi_{1,Z}\|\pi_{2,Z})], \tag{25}$$

and we arrive at the inequality as desired. $\qquad\square$

**Lemma 7.** *Let $d \in \Delta(S,A)$ be some state-action distribution, $\phi : S \times A \to Z$, and $\pi_Z : S \to \Delta(Z)$. Denote $\pi_{\alpha^*}$ as the optimal action decoder for $d, \phi$:*

$$\pi_{\alpha^*}(a|s,z) = \frac{d(s,a)\cdot\mathbb{1}[z=\phi(s,a)]}{\sum_{a'\in A}d(s,a')\cdot\mathbb{1}[z=\phi(s,a')]},$$

*and $\pi_{\alpha^*,Z}$ as the marginalization of $\pi_{\alpha^*}\circ\pi_Z$ onto $Z$:*

$$\pi_{\alpha^*,Z}(z|s) := \sum_{a\in A, z=\phi(s,a)}(\pi_{\alpha^*}\circ\pi_Z)(a|s) = \sum_{a\in A, z=\phi(s,a)}\sum_{\tilde{z}\in Z}\pi_{\alpha^*}(a|s,\tilde{z})\pi_Z(\tilde{z}|s).$$

*Then we have*

$$\pi_{\alpha^*,Z}(z|s) = \pi_Z(z|s) \tag{26}$$

*for all $z \in Z$ and $s \in S$.*

*Proof.*

$$\pi_{\alpha^*,Z}(z|s) = \sum_{a \in A, z = \phi(s,a)} \sum_{\tilde{z} \in Z} \pi_{\alpha^*}(a|s,\tilde{z}) \pi_Z(\tilde{z}|s) \tag{27}$$

$$= \sum_{a \in A, z = \phi(s,a)} \sum_{\tilde{z} \in Z} \frac{d(s,a) \cdot \mathbb{1}[\tilde{z} = \phi(s,a)]}{\sum_{a' \in A} d(s,a') \cdot \mathbb{1}[\tilde{z} = \phi(s,a')]} \pi_Z(\tilde{z}|s) \tag{28}$$

$$= \sum_{a \in A, z = \phi(s,a)} \frac{d(s,a) \cdot \mathbb{1}[z = \phi(s,a)]}{\sum_{a' \in A} d(s,a') \cdot \mathbb{1}[z = \phi(s,a')]} \pi_Z(z|s) \tag{29}$$

$$= \pi_Z(z|s) \sum_{a \in A, z = \phi(s,a)} \frac{d(s,a) \cdot \mathbb{1}[z = \phi(s,a)]}{\sum_{a' \in A} d(s,a') \cdot \mathbb{1}[z = \phi(s,a')]} \tag{30}$$

$$= \pi_Z(z|s), \tag{31}$$

and we have the desired equality. $\square$

**Lemma 8.** *Let $\pi_Z : S \to \Delta(Z)$ be a latent policy in $Z$ and $\pi_\alpha : S \times Z \to A$ be an action decoder, $\pi_{\alpha,Z}$ be the marginalization of $\pi_\alpha \circ \pi_Z$ onto $Z$:*

$$\pi_{\alpha,Z}(z|s) := \sum_{a \in A, z = \phi(s,a)} (\pi_\alpha \circ \pi_Z)(a|s) = \sum_{a \in A, z = \phi(s,a)} \sum_{\tilde{z} \in Z} \pi_\alpha(a|s,\tilde{z}) \pi_Z(\tilde{z}|s).$$

*Then for any $s \in S$ we have*

$$D_{\mathrm{TV}}(\pi_Z(s) \| \pi_{\alpha,Z}(s)) \le \max_{z \in Z} D_{\mathrm{TV}}(\pi_{\alpha^*}(s,z) \| \pi_\alpha(s,z)), \tag{32}$$

*where $\pi_{\alpha^*}$ is the optimal action decoder defined in Lemma 7 (and this holds for any choice of $d$ from Lemma 7).*

*Proof.*

$$D_{\mathrm{TV}}(\pi_Z(s) \| \pi_{\alpha,Z}(s)) \tag{33}$$

$$= \frac{1}{2} \sum_{z \in Z} |\pi_Z(z|s) - \pi_{\alpha,Z}(z|s)| \tag{34}$$

$$= \frac{1}{2} \sum_{z \in Z} \left| \pi_Z(z|s) - \sum_{a \in A, z = \phi(s,a)} \sum_{\tilde{z} \in Z} \pi_\alpha(a|s,\tilde{z}) \pi_Z(\tilde{z}|s) \right| \tag{35}$$

$$= \frac{1}{2} \sum_{z \in Z} \left| \pi_Z(z|s) - \sum_{a \in A, z = \phi(s,a)} \sum_{\tilde{z} \in Z} (\pi_\alpha(a|s,\tilde{z}) - \pi_{\alpha^*}(a|s,\tilde{z}) + \pi_{\alpha^*}(a|s,\tilde{z})) \pi_Z(\tilde{z}|s) \right| \tag{36}$$

$$= \frac{1}{2} \sum_{z \in Z} \left| \sum_{a \in A, z = \phi(s,a)} \sum_{\tilde{z} \in Z} (\pi_\alpha(a|s,\tilde{z}) - \pi_{\alpha^*}(a|s,\tilde{z})) \pi_Z(\tilde{z}|s) \right| \qquad \text{(by Lemma 7)} \tag{37}$$

$$\le \frac{1}{2} \mathbb{E}_{\tilde{z} \sim \pi_Z(s)} \left[ \sum_{z \in Z} \sum_{a \in A, z = \phi(s,a)} |\pi_\alpha(a|s,\tilde{z}) - \pi_{\alpha^*}(a|s,\tilde{z})| \right] \tag{38}$$

$$= \frac{1}{2} \mathbb{E}_{\tilde{z} \sim \pi_Z(s)} \left[ \sum_{a \in A} |\pi_\alpha(a|s,\tilde{z}) - \pi_{\alpha^*}(a|s,\tilde{z})| \right] \tag{39}$$

$$= \mathbb{E}_{\tilde{z} \sim \pi_Z(s)} [D_{\mathrm{TV}}(\pi_\alpha(s,\tilde{z}) \|, \pi_{\alpha^*}(s,\tilde{z}))] \tag{40}$$

$$\le \max_{z \in Z} D_{\mathrm{TV}}(\pi_\alpha(s,z) \| \pi_{\alpha^*}(s,z)), \tag{41}$$

$$\tag{42}$$

and we have the desired inequality. $\square$

**Lemma 9.** *Let $\pi_{1,Z}$ be the marginalization of $\pi_1$ onto $Z$ as defined in Lemma 6, and let $\pi_Z$, $\pi_\alpha$, $\pi_{\alpha,Z}$ be as defined in Lemma 8, and let $\pi_{\alpha^*,Z}$ be as defined in Lemma 7. For any $s \in S$ we have*

$$D_{\mathrm{TV}}(\pi_{1,Z}(s) \| \pi_{\alpha,Z}(s)) \le \max_{z \in Z} D_{\mathrm{TV}}(\pi_\alpha(s,z) \| \pi_{\alpha^*}(s,z)) + D_{\mathrm{TV}}(\pi_{1,Z}(s) \| \pi_Z(s)). \tag{43}$$

*Proof.* The desired inequality is achieved by plugging the inequality from Lemma 8 into the following triangle inequality:

$$D_{\mathrm{TV}}(\pi_{1,Z}(s)\|\pi_{\alpha,Z}(s)) \leq D_{\mathrm{TV}}(\pi_Z(s)\|\pi_{\alpha,Z}(s)) + D_{\mathrm{TV}}(\pi_{1,Z}(s)\|\pi_Z(s)). \tag{44}$$

$\square$

Our final lemma will be used to translate on-policy bounds to off-policy.

**Lemma 10.** *For two distributions* $\rho_1, \rho_2 \in \Delta(S)$ *with* $\rho_1(s) > 0 \Rightarrow \rho_2(s) > 0$, *we have,*

$$\mathbb{E}_{\rho_1}[h(s)] \leq (1 + D_{\chi^2}(\rho_1\|\rho_2)^{\frac{1}{2}})\sqrt{\mathbb{E}_{\rho_2}[h(s)^2]}. \tag{45}$$

*Proof.* The lemma is a straightforward consequence of Cauchy-Schwartz:

$$\mathbb{E}_{\rho_1}[h(s)] = \mathbb{E}_{\rho_2}[h(s)] + (\mathbb{E}_{\rho_1}[h(s)] - \mathbb{E}_{\rho_2}[h(s)]) \tag{46}$$

$$= \mathbb{E}_{\rho_2}[h(s)] + \sum_{s \in S} \frac{\rho_1(s) - \rho_2(s)}{\rho_2(s)^{\frac{1}{2}}} \cdot \rho_2(s)^{\frac{1}{2}} h(s) \tag{47}$$

$$\leq \mathbb{E}_{\rho_2}[h(s)] + \left(\sum_{s \in S} \frac{(\rho_1(s) - \rho_2(s))^2}{\rho_2(s)}\right)^{\frac{1}{2}} \cdot \left(\sum_{s \in S} \rho_2(s) h(s)^2\right)^{\frac{1}{2}} \tag{48}$$

$$= \mathbb{E}_{\rho_2}[h(s)] + D_{\chi^2}(\rho_1\|\rho_2)^{\frac{1}{2}} \cdot \sqrt{\mathbb{E}_{\rho_2}[h(s)^2]}. \tag{49}$$

Finally, to get the desired bound, we simply note that the concavity of the square-root function implies $\mathbb{E}_{\rho_2}[h(s)] \leq \mathbb{E}_{\rho_2}[\sqrt{h(s)^2}] \leq \sqrt{\mathbb{E}_{\rho_2}[h(s)^2]}$. $\square$

# B PROOFS FOR MAJOR THEOREMS

## B.1 PROOF OF THEOREM 1

*Proof.* Let $\pi_2 := \pi_\alpha \circ \pi_Z$, we have $\pi_{2,Z}(z|s) = \pi_{\alpha,Z}(z|s) = \sum_{a \in A, \phi(s,a) = z}(\pi_\alpha \circ \pi_Z)(z|s)$. By plugging the result of Lemma 9 into Lemma 6, we have

$$\mathrm{Err}_{d^{\pi_1}}(\pi_1, \pi_2, \overline{\mathcal{T}})] \leq \mathbb{E}_{s \sim d^{\pi_1}}\left[\max_{z \in Z} D_{\mathrm{TV}}(\pi_{\alpha^*}(s, z)\|\pi_\alpha(s, z)) + D_{\mathrm{TV}}(\pi_{1,Z}(s)\|\pi_Z(s))\right]. \tag{50}$$

By plugging this result into Lemma 5, we have

$$\mathrm{Err}_{d^{\pi_1}}(\pi_1, \pi_2, \mathcal{T}) \leq |A|\mathbb{E}_{(s,a) \sim (d^{\pi_1}, \mathrm{Unif}_A)}[D_{\mathrm{TV}}(\mathcal{T}(s, a)\|\overline{\mathcal{T}}(s, a))] \tag{51}$$

$$+ \mathbb{E}_{s \sim d^{\pi_1}}\left[\max_{z \in Z} D_{\mathrm{TV}}(\pi_{\alpha^*}(s, z)\|\pi_\alpha(s, z))\right] \tag{52}$$

$$+ \mathbb{E}_{s \sim d^{\pi_1}}[D_{\mathrm{TV}}(\pi_{1,Z}(s)\|\pi_Z(s))]. \tag{53}$$

By further plugging this result into Lemma 4 and let $\pi_1 = \pi_*$, we have:

$$\mathrm{Diff}(\pi_\alpha \circ \pi_Z, \pi_*) \leq \frac{\gamma|A|}{1 - \gamma} \cdot \mathbb{E}_{(s,a) \sim (d^{\pi_1}, \mathrm{Unif}_A)}[D_{\mathrm{TV}}(\mathcal{T}(s, a)\|\mathcal{T}_Z(s, \phi(s, a)))]$$

$$+ \frac{\gamma}{1 - \gamma} \cdot \mathbb{E}_{s \sim d^{\pi_*}}[\max_{z \in Z} D_{\mathrm{TV}}(\pi_{\alpha^*}(s, z)\|\pi_\alpha(s, z))]$$

$$+ \frac{\gamma}{1 - \gamma} \cdot \mathbb{E}_{s \sim d^{\pi_*}}[D_{\mathrm{TV}}(\pi_{*,Z}(s)\|\pi_Z(s))]. \tag{54}$$

Finally, by plugging in the off-policy results of Lemma 10 to the bound in Equation 54 and by applying Pinsker's inequality $D_{\mathrm{TV}}(\mathcal{T}(s, a)\|\mathcal{T}_Z(s, \phi(s, a)))^2 \leq \frac{1}{2}D_{\mathrm{KL}}(\mathcal{T}(s, a)\|\mathcal{T}_Z(s, \phi(s, a)))$, we have

$$\mathrm{Diff}(\pi_\alpha \circ \pi_Z, \pi_*) \leq C_1 \cdot \sqrt{\frac{1}{2}\underbrace{\mathbb{E}_{(s,a) \sim d^{\mathrm{off}}}[D_{\mathrm{KL}}(\mathcal{T}(s, a)\|\mathcal{T}_Z(s, \phi(s, a)))]}_{= J_{\mathrm{T}}(\mathcal{T}_Z, \phi)}}$$

$$+ C_2 \cdot \sqrt{\frac{1}{2}\underbrace{\mathbb{E}_{s \sim d^{\mathrm{off}}}[\max_{z \in Z} D_{\mathrm{KL}}(\pi_{\alpha^*}(s, z)\|\pi_\alpha(s, z))]}_{\approx \, \mathrm{const}(d^{\mathrm{off}}, \phi) + J_{\mathrm{DE}}(\pi_\alpha, \phi)}}$$

$$+ C_3 \cdot \sqrt{\frac{1}{2}\underbrace{\mathbb{E}_{s \sim d^{\pi_*}}[D_{\mathrm{KL}}(\pi_{*,Z}(s)\|\pi_Z(s))]}_{= \, \mathrm{const}(\pi_*, \phi) + J_{\mathrm{BC}, \phi}(\pi_Z)}}, \tag{55}$$

where $C_1 = \gamma|A|(1 - \gamma)^{-1}(1 + D_{\chi^2}(d^{\pi_*}\|d^{\text{off}})^{\frac{1}{2}})$, $C_2 = \gamma(1 - \gamma)^{-1}(1 + D_{\chi^2}(d^{\pi_*}\|d^{\text{off}})^{\frac{1}{2}})$, and $C_3 = \gamma(1 - \gamma)^{-1}$. Since the $\max_{z \in Z}$ is not tractable in practice, we approximate $\mathbb{E}_{s \sim d^{\text{off}}}[\max_{z \in Z} D_{\text{KL}}(\pi_{\alpha^*}(s, z)\|\pi_\alpha(s, z))]$ using $\mathbb{E}_{(s,a) \sim d^{\text{off}}}[D_{\text{KL}}(\pi_{\alpha^*}(s, \phi(s, a))\|\pi_\alpha(s, \phi(s, a)))]$, which reduces to $J_{\text{DE}}(\pi_\alpha, \phi)$ with additional constants. We now arrive at the desired off-policy bound in Theorem 1. $\qquad\square$

### B.2 PROOF OF THEOREM 2

**Lemma 11.** *Let $\rho \in \Delta(\{1, \ldots, k\})$ be a distribution with finite support. Let $\hat\rho_n$ denote the empirical estimate of $\rho$ from $n$ i.i.d. samples $X \sim \rho$. Then,*

$$\mathbb{E}_n[D_{\text{TV}}(\rho\|\hat\rho_n)] \leq \frac{1}{2} \cdot \frac{1}{\sqrt{n}} \sum_{i=1}^{k} \sqrt{\rho(i)} \leq \frac{1}{2} \cdot \sqrt{\frac{k}{n}}. \tag{56}$$

*Proof.* The first inequality is Lemma 8 in Berend & Kontorovich (2012) while the second inequality is due to the concavity of the square root function. $\qquad\square$

**Lemma 12.** *Let $\mathcal{D} := \{(s_i, a_i)\}_{i=1}^{n}$ be i.i.d. samples from a factored distribution $x(s, a) := \rho(s)\pi(a|s)$ for $\rho \in \Delta(S), \pi : S \to \Delta(A)$. Let $\hat\rho$ be the empirical estimate of $\rho$ in $\mathcal{D}$ and $\hat\pi$ be the empirical estimate of $\pi$ in $\mathcal{D}$. Then,*

$$\mathbb{E}_{\mathcal{D}}[\mathbb{E}_{s \sim \rho}[D_{\text{TV}}(\pi(s)\|\hat\pi(s))]] \leq \sqrt{\frac{|S||A|}{n}}. \tag{57}$$

*Proof.* Let $\hat{x}$ be the empirical estimate of $x$ in $\mathcal{D}$. We have,

$$\mathbb{E}_{s \sim \rho}[D_{\text{TV}}(\pi(s)\|\hat\pi(s))] = \frac{1}{2}\sum_{s,a}\rho(s) \cdot |\pi(a|s) - \hat\pi(a|s)| \tag{58}$$

$$= \frac{1}{2}\sum_{s,a}\rho(s) \cdot \left|\frac{x(s,a)}{\rho(s)} - \frac{\hat{x}(s,a)}{\hat\rho(s)}\right| \tag{59}$$

$$\leq \frac{1}{2}\sum_{s,a}\rho(s) \cdot \left|\frac{\hat{x}(s,a)}{\rho(s)} - \frac{\hat{x}(s,a)}{\hat\rho(s)}\right| + \frac{1}{2}\sum_{s,a}\rho(s) \cdot \left|\frac{\hat{x}(s,a)}{\rho(s)} - \frac{x(s,a)}{\rho(s)}\right| \tag{60}$$

$$= \frac{1}{2}\sum_{s,a}\rho(s) \cdot \left|\frac{\hat{x}(s,a)}{\rho(s)} - \frac{\hat{x}(s,a)}{\hat\rho(s)}\right| + D_{\text{TV}}(x\|\hat{x}) \tag{61}$$

$$= \frac{1}{2}\sum_{s}\rho(s) \cdot \left|\frac{1}{\rho(s)} - \frac{1}{\hat\rho(s)}\right|\left(\sum_{a}\hat{x}(s,a)\right) + D_{\text{TV}}(x\|\hat{x}) \tag{62}$$

$$= \frac{1}{2}\sum_{s}\rho(s) \cdot \left|\frac{1}{\rho(s)} - \frac{1}{\hat\rho(s)}\right| \cdot \hat\rho(s) + D_{\text{TV}}(x\|\hat{x}) \tag{63}$$

$$= D_{\text{TV}}(\rho\|\hat\rho) + D_{\text{TV}}(x\|\hat{x}). \tag{64}$$

Finally, the bound in the lemma is achieved by application of Lemma 11 to each of the TV divergences. $\qquad\square$

To prove Theorem 2, we first rewrite Theorem 1 as

$$\text{Diff}(\pi_Z, \pi_*) \leq (1)(\phi) + (2)(\phi) + C_3 \cdot \mathbb{E}_{s \sim d^{\pi_*}}[D_{\text{TV}}(\pi_{*,Z}(s)\|\pi_Z(s))], \tag{65}$$

where (1) and (2) are the first two terms in the bound of Theorem 1, and $C_3 = \frac{\gamma}{1-\gamma}$.

The result in Theorem 2 is then derived by setting $\phi = \phi_{\pi_{orcl}}$ and $\pi_Z := \pi_{\phi_{orcl},Z}$ and using the result of Lemma 12.

Note that the above sample analysis can be extended to the continuous latent action space characterized by Theorem 3 as follows.

**Theorem 13.** *Let $\phi_{orcl} := \mathcal{OPT}_\phi(\mathcal{D}^{\text{off}})$ and $\pi_{orcl,\theta}$ be the latent BC policy with respect to $\phi_{orcl}$. Let $d$ be the dimension of the continuous latent actions and $\|\phi\|_\infty$ be the $l_\infty$ norm of $\phi_{orcl}$ for any $s, a$. We have*

$$\mathbb{E}_{\mathcal{D}^{\pi_*}}[\text{Diff}(\pi_{\phi_{orcl},\theta}, \pi_*)] \leq (1)(\phi_{orcl}) + (2)(\phi_{orcl}) + C_4 \cdot d\|\phi\|_\infty\sqrt{\frac{2|S|}{n+1}},$$

*where (1), (2), and $C_4$ are the same as in Theorem 3.*

*Proof.* We use $\mu \in \mathbb{R}^{d \times |S|}$ to denote the optimal setting of $\theta$ which yields a zero $l_1$-norm of $\frac{\partial}{\partial \theta} \mathbb{E}_{s \sim d^\pi, a \sim \pi_*}[(\theta_s - \phi(s,a))^2]$; i.e.,

$$\mu_s = \mathbb{E}_{a \sim \pi_*(s)}[\phi(s,a)]. \tag{66}$$

According to Theorem 3, we want to bound the $l_1$-norm of $\frac{\partial}{\partial \theta} \mathbb{E}_{s \sim d^\pi, a \sim \pi_*}[(\theta_s - \phi(s,a))^2]$ evaluated at the approximate solution $\hat{\mu} \in \mathbb{R}^{d \times |S|}$ with respect to finite dataset $\mathcal{D}^{\pi_*}$; i.e.,

$$\hat{\mu}_s = \mathbb{E}_{a \sim \mathcal{D}^{\pi_*}(\cdot|s)}[\phi(s,a)], \tag{67}$$

with the convention that $\hat{\mu}_s = 0$ if $s$ does not appear in $\mathcal{D}^{\pi_*}$. To this end, we have the following derivation, which uses $\mathbb{E}_n$ to denote the expectation over realizations of $\hat{\mu}$ due to $n$-size draws of the target dataset $\mathcal{D}^{\pi_*}$:

$$\mathbb{E}_n \left[ \left\| \frac{\partial}{\partial \theta} \Big|_{\theta = \hat{\mu}} \mathbb{E}_{s \sim d^\pi, a \sim \pi_*} \left[ (\theta_s - \phi(s,a))^2 \right] \right\|_1 \right] = \mathbb{E}_n \left[ \mathbb{E}_{s \sim d^\pi} [\|\hat{\mu}_s - \mathbb{E}_{a \sim \pi_*}[\phi(s,a)]\|_1] \right] \tag{68}$$

$$= \mathbb{E}_n \left[ \mathbb{E}_{s \sim d^\pi} [\|\hat{\mu}_s - \mu_s\|_1] \right] \tag{69}$$

$$= \mathbb{E}_{s \sim d^\pi} \left[ \mathbb{E}_n [\|\hat{\mu}_s - \mu_s\|_1] \right]. \tag{70}$$

We now split up the inner expectation based on the number of times $k$ that $s$ appears in $\mathcal{D}^{\pi_*}$:

$$\mathbb{E}_{s \sim d^\pi} \left[ \mathbb{E}_n [\|\hat{\mu}_s - \mu_s\|_1] \right] = \mathbb{E}_{s \sim d^\pi} \left[ \sum_{k=0}^n \Pr[\text{count}(s) = k] \cdot \mathbb{E}_k [\|\hat{\mu}_s - \mu_s\|_1] \right] \tag{71}$$

$$\leq \sqrt{\mathbb{E}_{s \sim d^\pi} \left[ \sum_{k=0}^n \Pr[\text{count}(s) = k] \cdot \mathbb{E}_k [\|\hat{\mu}_s - \mu_s\|_1]^2 \right]} \tag{72}$$

$$\tag{73}$$

where $\mathbb{E}_k$ denotes the expectation over realizations of $\hat{\mu}_s$ over $k$-size draws of $a \sim \pi_*(s)$. By standard combinatorics, we know

$$\Pr[\text{count}(s) = k] = \binom{n}{k} d^\pi(s)^k (1 - d^\pi(s))^{n-k}. \tag{74}$$

Furthermore, for $k = 0$, we have

$$\mathbb{E}_k [\|\hat{\mu}_s - \mu_s\|_1]^2 = \|\mu_s\|_1^2 \leq d^2 \|\phi\|_\infty^2, \tag{75}$$

while for $k > 0$, since $\mathbb{E}_k[\hat{\mu}_s] = \mu_s$, we have

$$\mathbb{E}_k [\|\hat{\mu}_s - \mu_s\|_1]^2 \leq d \cdot \mathbb{E}_k [\|\hat{\mu}_s - \mu_s\|_2^2] = d \cdot \text{Var}_k [\hat{\mu}_s] \leq \frac{d^2 \|\phi\|_\infty^2}{k} \leq \frac{2 d^2 \|\phi\|_\infty^2}{k+1}. \tag{76}$$

Combining equations 74, 75, and 76 we have for any $k \geq 0$

$$d^\pi(s) \cdot \Pr[\text{count}(s) = k] \cdot \mathbb{E}_k [\|\hat{\mu}_s - \mu_s\|_1]^2 \leq \frac{2 d^2 \|\phi\|_\infty^2}{k+1} \binom{n}{k} d^\pi(s)^{k+1} (1 - d^\pi(s))^{n-k}$$

$$= \frac{2 d^2 \|\phi\|_\infty^2}{n+1} \binom{n+1}{k+1} d^\pi(s)^{k+1} (1 - d^\pi(s))^{n-k}, \tag{77}$$

and so by the binomial theorem,

$$\sum_{k=0}^n d^\pi(s) \cdot \Pr[\text{count}(s) = k] \cdot \mathbb{E}_k [\|\hat{\mu}_s - \mu_s\|_1]^2 \leq \frac{2 d^2 \|\phi\|_\infty^2}{n+1}. \tag{78}$$

Plugging the above into equation 72 we deduce

$$\mathbb{E}_{s \sim d^\pi} [\mathbb{E}_n [\|\hat{\mu}_s - \mu_s\|_1]] \leq d \|\phi\|_\infty \sqrt{\frac{2|S|}{n+1}}, \tag{79}$$

and we have the convergence rate as desired. $\qquad \square$

### B.3 PROOF OF THEOREM 3

*Proof.* The gradient term in Theorem 3 with respect to a specific column $\theta_s$ of $\theta$ may be expressed as

$$\frac{\partial}{\partial \theta_s} \mathbb{E}_{\tilde{s} \sim d^\pi, a \sim \pi(\tilde{s})}[(\theta_{\tilde{s}} - \phi(\tilde{s},a))^2]$$

$$= -2 \mathbb{E}_{a \sim \pi(s)}[d^\pi(s) \phi(s,a)] + 2 d^\pi(s) \theta_s$$

$$= -2 \mathbb{E}_{a \sim \pi(s)}[d^\pi(s) \phi(s,a)] + 2 \mathbb{E}_{z = \theta_s}[d^\pi(s) \cdot z], \tag{80}$$

and so,

$$w(s')^\top \frac{\partial}{\partial \theta_s} \mathbb{E}_{\tilde{s} \sim d^\pi, a \sim \pi(\tilde{s})}[(\theta_{\tilde{s}} - \phi(\tilde{s}, a))^2]$$
$$= -2\mathbb{E}_{a \sim \pi(s)}[d^\pi(s)\overline{\mathcal{T}}(s'|s, a)] + 2\mathbb{E}_{z=\theta_s}[d^\pi(s)w(s')^\top z]. \tag{81}$$

Summing over $s \in S$, we have:

$$\sum_{s \in S} w(s')^\top \frac{\partial}{\partial \theta_s} \mathbb{E}_{\tilde{s} \sim d^\pi, a \sim \pi(\tilde{s})}[(\theta_{\tilde{s}} - \phi(\tilde{s}, a))^2]$$
$$= 2\mathbb{E}_{s \sim d^\pi, a \sim \pi(s), z=\theta_s}[-\overline{\mathcal{T}}(s'|s, a) + \mathcal{T}_Z(s'|s, z)] \tag{82}$$

Thus, we have:

$$\mathrm{Err}_{d^\pi}(\pi, \pi_\theta, \overline{\mathcal{T}}) = \frac{1}{2} \sum_{s' \in S} \left| \mathbb{E}_{s \sim d^\pi, a \sim \pi(s), z=\theta_s}[-\overline{\mathcal{T}}(s'|s, a) + \mathcal{T}_Z(s'|s, z)] \right|$$
$$= \frac{1}{4} \sum_{s' \in S} \left| \sum_{s \in S} w(s')^\top \frac{\partial}{\partial \theta_s} \mathbb{E}_{\tilde{s} \sim d^\pi, a \sim \pi(\tilde{s})}[(\theta_{\tilde{s}} - \phi(\tilde{s}, a))^2] \right|$$
$$\leq \frac{1}{4} |S| \|w\|_\infty \cdot \left\| \frac{\partial}{\partial \theta} \mathbb{E}_{s \sim d^\pi, a \sim \pi(s)}[(\theta_s - \phi(s, a))^2] \right\|_1. \tag{83}$$

Then by combining Lemmas 4, 5, 10, and apply Equation 83 (as opposed to Lemma 6 as in the tabular case), we arrive at the desired bound in Theorem 3. □

## C  EXPERIMENT DETAILS

### C.1  ARCHITECTURE

We parametrize $\phi$ as a two-hidden layer fully connected neural network with 256 units per layer. A Swish (Ramachandran et al., 2017) activation function is applied to the output of each hidden layer. We use embedding size 64 for AntMaze and 256 for Ant and all DeepMind Control Suite (DMC) tasks after sweeping values of 64, 256, and 512, though we found TRAIL to be relatively robust to the latent dimension size as long as it is not too small (i.e., $\geq 64$). The latent skills in temporal skill extraction require a much smaller dimension size, e.g., 8 or 10 as reported by Ajay et al. (2020); Pertsch et al. (2021). We tried increasing the latent skill size for these work during evaluation, but found the reported value 8 to work the best. We additionally experimented with different extend of skill extraction, but found the previously reported $t = 10$ to also work the best. We implement the trajectory encoder in OPAL, SkiLD, and SPiRL using a bidirectional LSTM with hidden dimension 256. We use $\beta = 0.1$ for the KL regularization term in the $\beta$ VAE of OPAL (as reported). We also use 0.1 as the weight for SPiRL and SkiLD's KL divergence terms.

### C.2  TRAINING AND EVALUATION

During pretraining, we use the Adam optimizer with learning rate 0.0003 for 200k iterations with batch size 256 for all methods that require pretraining. During downstream behavioral cloning, learned action representations are fixed, but the action decoder is fine-tuned on the expert data as suggested by Ajay et al. (2020). Behavioral cloning for all methods including vanilla BC is trained with learning rate 0.0001 for 1M iterations. We experimented with learning rate decay of downstream BC by a factor of 3 at the 200k boundary for all methods. We found that when the expert sample size is small, decaying learning rate can prevent overfitting for all methods. The reported results are with learning rate decay on AntMaze and without learning rate decay on other environments for all methods. During the downstream behavioral cloning stage, we evaluate the latent policy combined with the action decoder every 10k steps by executing $\pi_\alpha \circ \pi_Z$ in the environment for 10 episodes and compute the average total return. Each method is run with 4 seeds where each seed corresponds to one set of action representations and downstream imitation learning result on that set of representations. We report the mean and standard error for all methods in the bar and line figures.

### C.3  MODIFICATION TO SKILD AND SPIRL

Since SkiLD (Pertsch et al., 2021) and SPiRL (Pertsch et al., 2020) are originally designed for RL as opposed to imitation learning, we replace the downstream RL algorithms of SkiLD and SPiRL by behavioral cloning with regularization (but keep skill extraction the same as the original methods). Specifically, for SkILD, we apply a KL regularization term between the latent policy and the learned

skill prior in the suboptimal offline dataset during pretraining, and another KL regularization term between the latent policy and a learn "skill posterior" on the expert data as done in the original paper during downstream behavioral cloning. We do not need to train the binary classifier that SkiLD trains to decide which regularizer to apply because we know which set of actions are expert versus suboptimal in the imitation learning setting. For SPiRL, we apply the KL divergence between latent policy and skill prior extracted from offline data (i.e., using the red term in Algorithm 1 of Pertsch et al. (2020)) as an additional term to latent behavioral cloning.

### C.4 DATASET DETAILS

**AntMaze.** For the expert data in AntMaze, we use the goal-reaching expert policies trained by Ajay et al. (2020) (expert means that the agent is trained to navigate from the one corner of the maze to the opposite corner) to collect $n = 10$ trajectories. For the suboptimal data in AntMaze, we use the full D4RL datasets `antmaze-large-diverse-v0`, `antmaze-medium-play-v0`, `antmaze-medium-diverse-v0`, and `antmaze-medium-play-v0`.

**Ant.** For the expert data in Ant, we use a small set of expert trajectories selected by taking either the first 10k or 25k transitions from `ant-expert-v0` in D4RL, corresponding to about 10 and 25 expert trajectories, respectively. For the suboptimal data in Ant, we use the full D4RL datasets `ant-medium-v0`, `ant-medium-replay-v0`, and `ant-random-v0`.

**RL Unplugged.** For DeepMind Control Suite (Tassa et al., 2018) set of tasks, we use the RL Unplugged (Gulcehre et al., 2020) dataset. For the expert data, we take $\frac{1}{10}$ of the trajectories whose episodic reward is among the top $20\%$ of the open source RL Unplugged datasets following the setup in Zolna et al. (2020). For the suboptimal data, we use the bottom $80\%$ of the RL Unplugged dataset. Table 1 records the total number of trajectories available in RL Unplugged for each task ($80\%$ of which are used as suboptimal data), and the number of expert trajectories used in our evaluation.

| Task | # Total | # $\mathcal{D}^{\pi_*}$ |
|---|---|---|
| cartpole-swingup | 40 | 2 |
| cheetah-run | 300 | 3 |
| fish-swim | 200 | 1 |
| humanoid-run | 3000 | 53 |
| walker-stand | 200 | 4 |
| walker-walk | 200 | 6 |

Table 1: Total number of trajectories from RL Unplugged (Gulcehre et al., 2020) locomotion tasks used to train CRR (Wang et al., 2020) and the number of expert trajectories used to train TRAIL. The bottom $80\%$ of # Total is used to learn action representations by TRAIL.

# D ADDITIONAL EMPIRICAL RESTULS

## D.1 ADDITIONAL BASELINES FOR RL UNPLUGGED

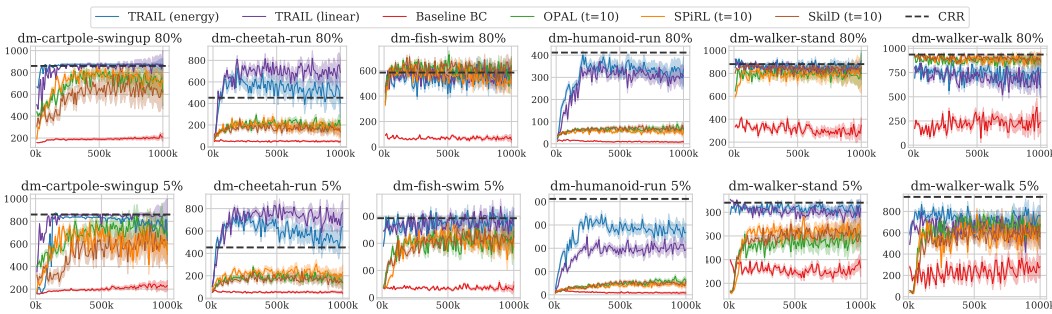

Figure 6: Average task rewards (over 4 seeds) of TRAIL EBM (Theorem 1), TRAIL linear (Theorem 3), and OPAL, SkiLD, SPiRL trained on the bottom 80% (top) and bottom 5% (bottom) of the RL Unplugged datasets followed by behavioral cloning in the latent action space. Baseline BC achieves low rewards due to the small expert sample size. Dotted lines denote the performance of CRR (Wang et al., 2020) trained on the full dataset with reward labels.

## D.2 FRANKAKITCHEN RESULTS

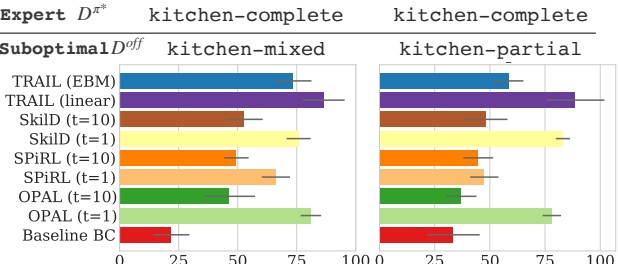

Figure 7: Average rewards (over 4 seeds) of TRAIL EBM (Theorem 1), TRAIL linear (Theorem 3), and baseline methods pretrained on `kitchen-mixed` and `kitchen-partial` from D4RL to imitate `kitchen-complete`. TRAIL linear without temporal abstraction performs slightly better than SKiLD and OPAL with temporal abstraction over 10 steps.

### D.3 DISCRETE MAZE RESULTS

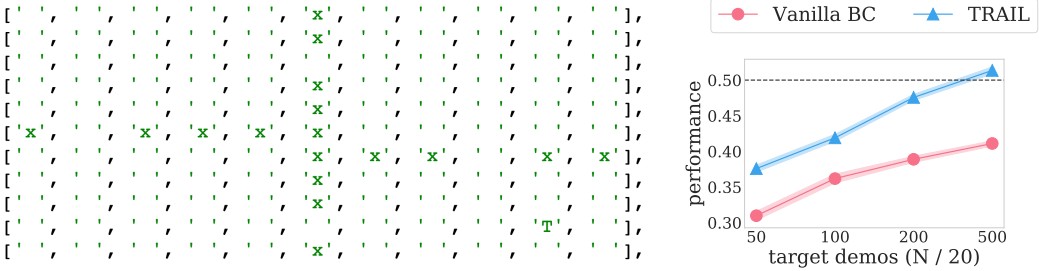

Figure 8: Average task rewards (over 4 seeds) of TRAIL EBM (Theorem 1) and vanilla BC (right) in a discrete four-room maze environment (left) where an agent is randomly placed in the maze and tries to reach the target 'T'. TRAIL learns a discrete latent action space of size 4 from the discrete original action space of size 12 on 500 uniform random trajectories of length 20 shows clear benefit over vanilla BC on expert data.

We conduct additional evaluation on an environment with tabular state and action spaces. As shown in Figure 8, an agent is randomly placed into a four-room environment, and the task is to navigate to the target 'T'. The task reward is 1 at 'T' and 0 elsewhere. There are 12 discrete actions corresponding to rotating clockwise by $90, 180, 270, 360$ degrees, rotating counterclockwise by $90, 180, 270, 360$ degrees, moving forward by 1 or 2 grids, and moving backward by 1 or 2 grids (the action space is artificially blown up as suggested by the reviewer). TRAIL is pretrained on 500 trajectories of length 20 with uniform action selection. The expert demonstration always navigates to the target 'T' from any random starting location. TRAIL's latent action dimension is set to 4. We see that TRAIL with a smaller latent action space offers benefits over vanilla BC.

# E    ABLATION STUDY

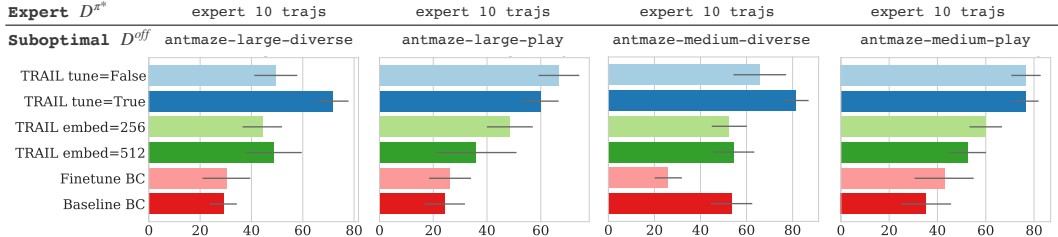

Figure 9: Ablation study on action decoder finetuning, latent dimension size, and pretraining baseline BC on suboptimal data in the AntMaze environment. TRAIL with default embedding dimension 64 and finetuning the action decoder corresponds to the second row. Other dimension size (256 and 512) lead to worse performance. Finetuning the action decoder on the expert data has some small benefits. Pretraining BC on suboptimal data before finetuning on expert does not lead to significantly better performance.

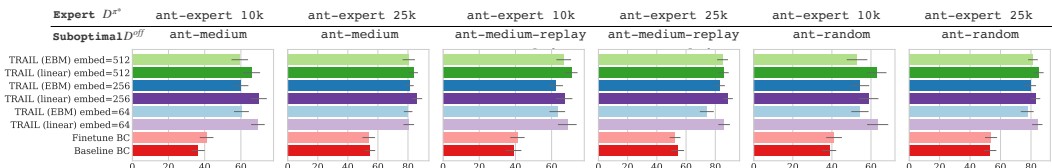

Figure 10: Ablation study on latent dimension size in the Ant environment. TRAIL is generally robust to the choices of the latent action dimension (64, 256, 512) for the Ant task.

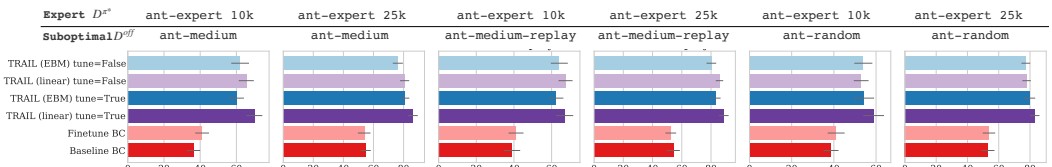

Figure 11: Ablation study on finetuning the action decoder in the Ant environment. Finetuning the action decoder leads to a slight benefit.

## F  VISUALIZATION OF LATENT ACTIONS

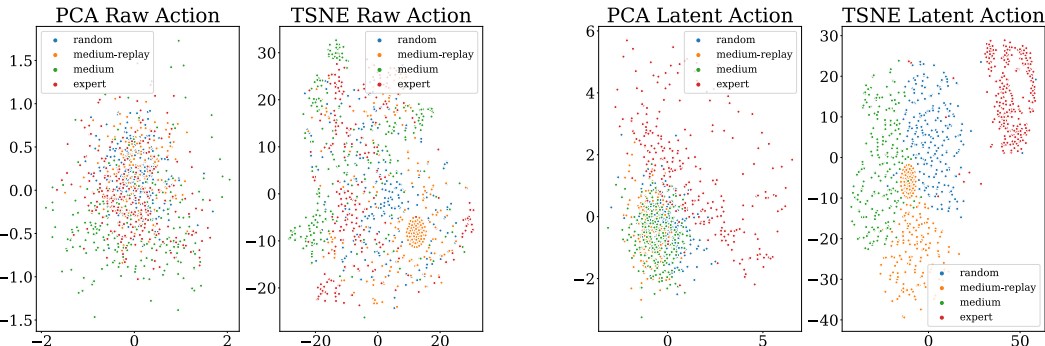

Figure 12: PCA and t-SNE visualizations of the `random`, `medium-replay`, `medium`, and `expert` D4RL Ant datasets. Without action representation learning (left), the distinction between expert and suboptimal actions is not obvious. The latent actions of TRAIL (right), on the other hand, results in the expert latent actions being more visually separable from suboptimal actions.

