# OpenReview forum: "TRAIL: Near-Optimal Imitation Learning with Suboptimal Data"
_ICLR.cc/2022/Conference — ICLR 2022 Poster_

### Official Review · Reviewer_xchs · 2021-10-29

**Correctness:** 4
**Technical Novelty And Significance:** 3
**Empirical Novelty And Significance:** 3
**Recommendation:** 6
**Confidence:** 3

**Main Review:**

# Strengths:

(1) The method is tested in several challenging environments. In all of them, TRAIL performs better than or is on par with the baselines.

(2) The paper is well written, and the theory appears to be sound

# Weaknesses:

(1) The bound of the imitation learning error derived in theorem 1 and 2 relies on the fact that $|Z| < |A|$. In all of the experiments, the action embedding ($> 64$) is bigger than the original action space ($=8$ for the ant). Could the authors provide an intuition of why such big action embeddings are required? This is somewhat contradictory to the theory.

(2) The comparison to the offline reinforcement learning setting is interesting.
However, the big promise of offline RL is to extract an optimal policy of the MDP induced by the offline dataset. That means, in principle, offline RL should be able to recover an optimal policy just from random data (given a sufficient state and action space coverage in the offline dataset) or a multitude of behaviors.
I would like to see a small paragraph in section 5.3 discussing this setting and how this affects BC.

(3) In C.2, the authors write that the action decoder was fine-tuned on the expert data while keeping the action representations fixed. How does that affect the final performance?

# Needs clarification and general comments

(1) The authors claim that “[a] good action representation should enable good imitation learning.”
Could the authors give an intuition (independent of the empirical results) on why a good BC action representation should emerge from a good transition-based representation?

(2) In theorem 1 $\pi_{Z}$ is defined as $\pi_{Z} \colon S \times Z \to \Delta(A)$. If I am not mistaken, that should be $\pi_{Z} \colon S \to Z$

(3) I think in the sample complexity paragraph after (1) and (2) $J_{T}$ and $J_{DE}$ are missing, respectively.

(4) In equation 5 $\psi(s^{\prime})$ appears for the first time. However, I am failing the find the definition of $\psi(s)$

(5) The BC baseline is only trained with the expert data. Is there a way of incorporating the suboptimal data in the training of the vanilla BC baseline? Maybe by first training with the suboptimal data and later fine-tuning on the expert data.

(6) I would like to see a plot showing the effects of the different design choices (size of the latent embedding, fine-tuning the action encoder) on the performance.

(7) To empirically prove the theoretical claims, the authors could run another experiment in which they artificially blow up the action dimensionality and check if they can learn a much smaller embedding.


**Summary Of The Paper:**

The paper proposes a method to accelerate behavioral cloning (BC) (especially in the low data regime) by utilizing a (much larger) auxiliary dataset of suboptimal behaviors.
The authors claim that learning a good latent action representation (in this case, by learning a transition-based action representation from the auxiliary dataset) should make the imitation learning problem easier.
The training is divided into two phases: (1) learning a factorized transition model in conjunction with an action embedding and action decoder, (2) learning a latent policy via the BC objective hoping that the transition-based representation accelerates the BC learning.

**Summary Of The Review:**

At the moment, I tend towards a weak reject because I don’t find the connection between theory and empirical evaluation too convincing. The reduction in the BC error shown in theorem 1 and 2 are based on the assumption that $|Z| < |A|$. However, if I am not mistaken, that is not the case in any of the experiments where $|Z|$ is usually much bigger than $|A|$.
On the other side, the empirical results show a clear improvement over the baselines and vanilla BC, indicating that the learned latent embedding is helpful.
I am willing to change my score if the authors strengthen the connection between the theory and the empirical results. For instance, they could explain why the transition-based representation might help BC, although the latent embedding is bigger than the original action space.

### Decision due to rebuttal

The authors addressed all of my concerns sufficiently well. Therefore I am happy to raise my score from a 5 to a 6.

---

> ### Author Response · Authors · 2021-11-16
> **Author Response (1/2)**
>
> Thank you for the detailed review, which we address below and in the updated paper.
>
> > (1) the action embedding (>64) is bigger than the original action space (=8 for the ant).
>
> The sample analysis with $|Z| < |A|$ in our theory is for tabular actions where $|Z|$ and $|A|$ denote the number of discrete actions as opposed to the number of dimensions in the continuous action space. Sample complexity depends on the number of parameters, which are $|Z|$ and $|A|$ in the tabular case but depends on the complexity of the function class in continuous action space with nonlinear function approximation. For instance, the expert policy in the raw action space could have been a complex multimodal mixture of Gaussian, so even if the raw action dimension is smaller, we still need a higher dimensional deterministic latent policy to recover the expert policy.
>
> Since all of our initial experiments were in continuous state-action space with neural network approximated $\phi$, the optimization was easier with larger latent dimension size. To better support our theory in tabular state-action space, we include an additional discrete maze experiment in Appendix D to show that lower-dimensional tabular $Z$ does offer benefits over vanilla BC in tabular $A$. In this experiment ([Figure 8](https://i.ibb.co/qp6KcgC/pointmaze.jpg)) an agent is randomly placed into a four-room environment. There are 12 discrete actions corresponding to rotating clockwise by 90, 180, 270, 360 degrees, rotating counterclockwise by 90, 180, 270, 360 degrees, moving forward by 1 or 2 grids, and moving backward by 1 or 2 grids (the action space is artificially blown up as suggested by the reviewer). TRAIL is pretrained on 500 trajectories of length 20 with uniform action selection. The expert demonstration always navigates to the target ‘T’. TRAIL’s latent action dimension is set to 4. We see that TRAIL with a smaller latent action space offers benefits over vanilla BC.
>
> > (2) Comparison and comments on offline RL
>
> First of all, we are not claiming that TRAIL outperforms offline RL; we are demonstrating an alternative to how suboptimal offline data could be used, especially when there is no reward signal. Secondly, the reviewer’s claim about "in principle, offline RL should be able to recover an optimal policy just from random data given sufficient coverage" is not necessarily true [1]. Lastly, there has been empirical evidence showing that offline RL can underperform BC [2].
>
> [1] Zanette, Andrea. "Exponential lower bounds for batch reinforcement learning: Batch rl can be exponentially harder than online rl." International Conference on Machine Learning. PMLR, 2021.
>
> [2] Florence, Pete, et al. "Implicit behavioral cloning." arXiv preprint arXiv:2109.00137 (2021).
>
> > (3) The effect of finetuning the action decoder
>
> We include additional ablation in Appendix E to show the effect of finetuning the action decoder ([Figure 9](https://i.ibb.co/wL5Gv80/antmaze-ablation.jpg) and [Figure 11](https://i.ibb.co/ZB8KK25/ant-ablate-tune.jpg)). Finetuning the action decoder leads to a slight performance improvement.
>
> > (1) Intuition on why a good BC action representation should emerge from a good transition-based representation
>
> The raw action of turning clockwise by 90 degrees and turning counterclockwise by 270 degrees (as well as the low-level motor control mechanisms involved) can be captured by a latent action, e.g., face east. Downstream imitation only needs to decide on the simpler latent action e.g., where a robot should face and move to; the exact low-level mechanisms of motor controls are handled by the action decoder (pretrained on offline data).
>
> > (2) Theorem 1 definition of $\pi_Z$
>
> Yes this is a typo and should be $\pi_Z:S\to Z$.
>
> > (3) I think in the sample complexity paragraph after (1) and (2) $J_T$ and $J_{DE}$ are missing, respectively.
>
> $J_T$ and $J_{DE}$ are defined to simplify how the objective is presented. The sample complexity bound should be in terms of the error in the full expression of (1) and (2).
>
> > (4) The definition of $\psi(s′)$ in Equation 5
>
> Thank you for pointing this out. $\psi: S \to Z$ is just a function of $s'$ that is used only during pretraining. We added this definition to the paper.
>
> > (5) Is there a way of incorporating the suboptimal data in the training of the vanilla BC baseline? Maybe by first training with the suboptimal data and later fine-tuning on the expert data.
>
> We include this version of BC trained on suboptimal data and finetuned on expert data as suggested by the reviewer in Appendix E ([Figure 9](https://i.ibb.co/wL5Gv80/antmaze-ablation.jpg), [Figure 10](https://i.ibb.co/6Hzdkq4/ant-ablate-dim.jpg), and [Figure 11](https://i.ibb.co/ZB8KK25/ant-ablate-tune.jpg)). In general, this version of BC performs similarly to vanilla BC trained only on expert data.
>
> CONTINUED

---

> > ### Author Response · Authors · 2021-11-16
> > **Author Response (2/2)**
> >
> > > (6) Size of the latent embedding, fine-tuning the action encoder) on the performance.
> >
> > We include additional ablation in Appendix E to show the effect of size of the latent embedding and finetuning the action decoder ([Figure 9](https://i.ibb.co/wL5Gv80/antmaze-ablation.jpg), [Figure 10](https://i.ibb.co/6Hzdkq4/ant-ablate-dim.jpg), and [Figure 11](https://i.ibb.co/ZB8KK25/ant-ablate-tune.jpg)). We found that 64 works the best among [64, 256, 512] for AntMaze, and 256 works the best for Ant. Finetuning the action decoder leads to a slight performance improvement.
> >
> > > (7) Artificially blow up the action dimensionality and check if they can learn a much smaller embedding
> >
> > As described in our response to weakness (1), we include an additional discrete maze experiment in Appendix D to show that lower-dimensional tabular Z does offer benefits over vanilla BC in tabular A when the discrete action space is artificially blown up as suggested by the reviewer. ([Figure 8](https://i.ibb.co/qp6KcgC/pointmaze.jpg)).

---

> > ### Comment · Reviewer_xchs · 2021-11-19
> > **Comments on the author responds**
> >
> > Thank you for the detailed response.
> >
> > > The sample analysis with  in our theory is for tabular actions where  and  denote the number of discrete actions as opposed to the number of dimensions in the continuous action space. Sample complexity depends on the number of parameters, which are  and  in the tabular case but depends on the complexity of the function class in continuous action space with nonlinear function approximation. For instance, the expert policy in the raw action space could have been a complex multimodal mixture of Gaussian, so even if the raw action dimension is smaller, we still need a higher dimensional deterministic latent policy to recover the expert policy.
> >
> > > Since all of our initial experiments were in continuous state-action space with neural network approximated , the optimization was easier with larger latent dimension size. To better support our theory in tabular state-action space, we include an additional discrete maze experiment in Appendix D to show that lower-dimensional tabular  does offer benefits over vanilla BC in tabular . In this experiment (Figure 8) an agent is randomly placed into a four-room environment. There are 12 discrete actions corresponding to rotating clockwise by 90, 180, 270, 360 degrees, rotating counterclockwise by 90, 180, 270, 360 degrees, moving forward by 1 or 2 grids, and moving backward by 1 or 2 grids (the action space is artificially blown up as suggested by the reviewer). TRAIL is pretrained on 500 trajectories of length 20 with uniform action selection. The expert demonstration always navigates to the target ‘T’. TRAIL’s latent action dimension is set to 4. We see that TRAIL with a smaller latent action space offers benefits over vanilla BC.
> >
> > That makes sense. Thanks for the clarification.
> >
> > > We include additional ablation in Appendix E to show the effect of finetuning the action decoder (Figure 9 and Figure 11). Finetuning the action decoder leads to a slight performance improvement.
> >
> > Great, thanks.
> >
> > > The raw action of turning clockwise by 90 degrees and turning counterclockwise by 270 degrees (as well as the low-level motor control mechanisms involved) can be captured by a latent action, e.g., face east. Downstream imitation only needs to decide on the simpler latent action e.g., where a robot should face and move to; the exact low-level mechanisms of motor controls are handled by the action decoder (pretrained on offline data).
> >
> > I see. Thanks for providing this intuition.
> >
> > > We include this version of BC trained on suboptimal data and finetuned on expert data as suggested by the reviewer in Appendix E (Figure 9, Figure 10, and Figure 11). In general, this version of BC performs similarly to vanilla BC trained only on expert data.
> >
> > Great
> >
> > > As described in our response to weakness (1), we include an additional discrete maze experiment in Appendix D to show that lower-dimensional tabular Z does offer benefits over vanilla BC in tabular A when the discrete action space is artificially blown up as suggested by the reviewer. (Figure 8).
> >
> > Interesting. Thanks for this additional experiment.

---

### Official Review · Reviewer_M6PX · 2021-11-02

**Correctness:** 3
**Technical Novelty And Significance:** 3
**Empirical Novelty And Significance:** 3
**Recommendation:** 8
**Confidence:** 3

**Main Review:**

## Strengths

- The paper proposes a theoretically justified method to learn from expert and sub-optimal data. The method is quite flexible in the sense that it requires a minimal assumption on the sub-optimality of the data (i.e., only that sub-optimal data should have an overlapped support with the expert data). Though, the sample complexity result relies on the assumption of the optimal action encoder which may not be realizable in practice.

- The experiments are done thoroughly to evaluate the efficiently and robustness of TRAIL. A sufficient number of baselines is compared against TRAIL in a sufficient number of benchmark tasks. Still, the results could be made stronger by considering more practical tasks such as robot arm manipulation.

## Weakness

- The main issue is on the clarity of the paper. In particular, I do not understand $\theta_s$ and $\theta$ is section 4.2. Is $\theta_s$ the $s$-th column of the matrix $\theta \in \mathbb{R}^{d \times |S|}$? If so, is $\theta : S \mapsto \mathbb{R}^d$ a typo or you assume it is a linear map? I also do not fully understand about the linear transition model in section 4.3 and the cosine non-linearity. Why is the cosine applied here when linearity is the goal?

## questions
- How are the number and dimensionality of latent action chosen for TRAIL EBM and TRAIL linear in each task? Are they tuned differently for each task or the same value is used across tasks?


**Summary Of The Paper:**

The paper considers an imitation learning (IL) problem with both expert and suboptimal demonstrations. The paper claims that sub-optimal demonstrations can be used to learn latent action abstractions which can improve the efficiency of down-stream IL. To solve this problem, the paper proposes TRAIL, which pre-trains an action encoder-decoder and a latent transition model using sub-optimal data, and performs behavioral cloning (BC) with expert data to learn a policy in the latent action space. The paper derives error bounds showing that the pre-training step and the down-stream BC step contribute to solving an IL problem in the original space from the view of divergence minimization. The paper also derives a bound showing that an optimal action abstraction may improve the sample complexity of BC. Experiments  indicate that TRAIL effectively learns from a limited number of expert demonstrations and is robust to the sub-optimality of sub-optimal demonstrations.

Main contributions: An effective method to learn an expert policy from few expert data and a large set of sub-optimal data. The method is theoretically justified and empirically well-supported.


**Summary Of The Review:**

I find the problem setting and proposed method very interesting. The theoretical analyses are valuable, and the experiments are convincing. Though, there is an issue regarding the clarity which make it difficult to fully understand the paper. I rate the paper as above the borderline.

** Update after authors' rebuttal **
The rebuttal and the revision address my concerns on the paper. I think more positive about the paper and have increased the score. The new result (Figure 7) in the appendix is a good addition and I suggest including it in the main paper as well.

---

> ### Author Response · Authors · 2021-11-16
> **Author Response**
>
> Thank you for the review. We improved clarity in our sample analysis and notations in our theorem, and included more details about random Fourier features. We also included additional experiments on robot manipulation and ablation as suggested by the reviewer.
>
> > Though, the sample complexity result relies on the assumption of the optimal action encoder which may not be realizable in practice.
>
> We are not making this optimality assumption in our sample complexity analysis. The oracle latent action representation function does not need to be optimal; our error terms should take into account any errors in $\phi$ or $\pi_\alpha$. We realized this confusion was caused by our notation, and we modified $\phi_{opt}$ to $\phi_{orcl}$ (oracle as opposed to optimal) in Theorem 2.
>
> > Considering more practical tasks such as robot arm manipulation
>
> We conducted additional experiments on FrankaKitchen from D4RL in appendix D ([Figure 7](https://i.ibb.co/mvqWTHN/kitchen.jpg)). In these experiments, the suboptimal offline data are the full kitchen-fixed-v0 or kitchen-partial-v0 (\~130k samples), and the expert demonstration is the kitchen-complete-v0 (\~3k samples). We see that TRAIL (linear) without temporal abstraction performs similarly to other method with temporal abstraction on this manipulation task.
>
> > Is $\theta: S↦R^d$ a typo or you assume it is a linear map?
>
> Thanks for pointing out this typo. $\theta_s$ is indeed the $s$-th column of the matrix $\theta\in R^{d\times|S|}$. The text should have been "there always exists a deterministic policy $\pi_\theta: S↦R^d$.
>
> > Why is the cosine applied here when linearity is the goal?
>
> The random Fourier features from the kernel literature [1] suggests that, for any k-dimensional vectors x and y, we can approximate $e^{−\|x − y\|^2/2} \approx \frac{d}{2} \varphi(x)^\top \varphi(y)$, for some $\varphi(x) = \cos(Wx + b)$, where $W$ is a $d\times k$ matrix with entries sampled from a unit Gaussian and $b$ a vector with entries sampled uniformly from $[0, 2\pi]$, hence achieving the linearity as desired.
>
> > Dimensionality of the latent action.
>
> We did a sweep over 64, 256, 512 for the dimension size (originally described in Appendix C1), and ended up using 64 for AntMaze and 256 for the other tasks. We now include plots of this ablation in Appendix E ([Figure 9](https://i.ibb.co/wL5Gv80/antmaze-ablation.jpg) and [Figure 10](https://i.ibb.co/6Hzdkq4/ant-ablate-dim.jpg)).
>
> [1] Rahimi, Ali, and Benjamin Recht. "Random Features for Large-Scale Kernel Machines." NIPS. Vol. 3. No. 4. 2007.

---

> > ### Author Response · Authors · 2021-11-19
> > **Author Follow-up**
> >
> > Dear Reviewer, we would like to ask if your concerns around the clarity of our method have been addressed, or if there were any other issues that would prevent you from increasing your score. Please let us know, and thank you for your time.

---

> > > ### Comment · Reviewer_M6PX · 2021-11-22
> > > **Response**
> > >
> > > Thank you for the response. The revision addresses my concerns on the paper. I will update the review after a reviewer discussion period.

---

### Official Review · Reviewer_nwvX · 2021-11-03

**Correctness:** 3
**Technical Novelty And Significance:** 3
**Empirical Novelty And Significance:** 2
**Recommendation:** 6
**Confidence:** 3

**Main Review:**

### Strength ###
- This paper considers a practical imitation learning setting in which there’s a very small amount of high-quality demonstrations that are expensive to obtain and there’s a large amount of (task-independent) suboptimal data that is cheaper to get.
- The proposed algorithm is simple and theoretically-grounded.
- This method of learning action reparameterization is applicable beyond imitation learning. Potential applications include offline reinforcement learning and transfer learning.

### Suggestions and Questions ###
- In the analysis, the paper considers the discrete state (and discrete action) setting. Is it possible to extend the analysis to continuous spaces?
  - In Theorem 3, when the latent action space is continuous, how does \pi_{alpha^*} in the second error term adapt from the first equation in page 5 which requires the latent action space to be discrete?
  - Sample complexity is studied under the tabular case in page 5. What’s the sample complexity (e.g. Theorem 2) when latent action space is continuous?
- In the TRAIL EBM for Theorem 1 paragraph in Section 4.3, you wrote “We allow \phi to be updated while optimizing J_{DE}”. Would this increase the first error term (Eq. (1))? Have you tried to optimize \phi, T, and \pi_\alpha simultaneously?
- I am excited about the experimental results. They look very promising.
  - However, It’s not clear to me whether learning action reparameterization is superior to learning state encoding and learning transition models without factorization. For example, a naive baseline may be interesting to consider is learning a transition model from the large amount of suboptimal data and then applying GAIL or AIRL to match the state distribution of the small amount of demonstrations using the learned transition model.
  - Some form of visualizing the latent action space learned may be interesting.

### Other Comments ###
- \pi_k in the second equation in page 5, \pi_5 has not been introduced
- In Theorem 1. Is \pi_Z S -> \Delta(Z)?
- In Section 4.3 TRAIL EBM for Theorem 1, “we set \rho to be the distribution of s’ in d^{off}”, does that mean you sample both the samples in the contrastive loss from the suboptimal data?

### Post-rebuttal comments ###
Thank the authors very much for their response in a short amount of time. All of my questions have been clarified. As before, I am leaning towards accepting the paper.


**Summary Of The Paper:**

The paper proposes an imitation learning algorithm, TRAIL, that can benefit from a large amount of suboptimal demonstrations besides a small amount of high-quality demonstrations. This is achieved through learning a factored transition model with action reparameterization from the suboptimal or even random demonstrations before doing behavior cloning. The authors analyze the error bound of the algorithm and show improved sample complexity with action reparameterization. Moreover, TRAIL is verified on a set of navigation and locomotion imitation learning tasks and it outperforms baselines based on temporal action abstraction in terms of task success rate.

**Summary Of The Review:**

Because of the strengths discussed above, I am leaning towards accepting the paper.

---

> ### Author Response · Authors · 2021-11-16
> **Author Response**
>
> Thank you for the detailed suggestions, which we address below and in the updated paper by providing sample analysis for continuous action space, visualization of the latent action space, and additional clarifications.
>
> > In the analysis, the paper considers the discrete state (and discrete action) setting. Is it possible to extend the analysis to continuous spaces?
>
> The discrete setting is only for the sample complexity analysis; everything else in our theorems should apply to general MDPs. When the original action space is continuous, the denominator of $\pi_{\alpha^*}$ in the first equation of page 5 would involve an integration over the action space as opposed to a sum. Note that this integration (or sum) only shows up in the bounds and is not being optimized (reduced to a constant).
>
> We have included additional sample complexity analysis below for continuous actions (See Theorem 13 and proof in the Appendix):
>
> $E_{\mathcal{D}}[D_{TV}(\pi_{\phi_{\text{orcl}},\theta}, \pi^*)] \le (1)(\phi_{\text{orcl}}) + (2)(\phi_{\text{orcl}}) + C_4\cdot||\phi||_\infty\sqrt{\frac{2d|S|}{n+1}}$
>
>
> > Would allowing $\phi$ to be updated while optimizing $J_{DE}$ increase the first error term (Eq. (1))? Have you tried to optimize $\phi$, $T$, and $\pi_\alpha$ simultaneously?
>
> We note that in our experiments $J_{DE}$ is not updated separately from the first error term in Eq 1. Rather, $\phi$, $T_Z$, and $\pi_\alpha$ are all optimized simultaneously on $D_{\text{off}}$. Moreover, we highlight that these two terms are not necessarily in conflict with each other; in fact, the optimal $\phi$ should achieve 0 error in both (1) and (2). We have tried some experiments not allowing the gradient of (2) to affect $\phi$, but these generally made the performance worse.
>
> > Learning a transition model from the large amount of suboptimal data and then applying GAIL
>
> The focus of this paper is to learn a latent action space. For this reason, our theoretical and empirical comparisons are predominantly with respect to existing works that use offline data in this way: to learn latent action spaces. Other mechanisms for using offline data are numerous -- model-based offline RL, state representation learning, algorithms like ValueDICE, GAIL with a learned dynamics model, etc. TRAIL solely focuses on using the offline data for latent action space learning (which is theoretically motivated), but the question of what is the ideal approach for leveraging offline data is still open (and very interesting).
>
> > Visualization of the latent action space
>
> We include t-SNE and PCA visualizations of the latent actions extracted by TRAIL on random, medium-replay, medium, and expert datasets from the Ant task in Appendix F ([Figure 12](https://i.ibb.co/2MP4P3Z/visualization.jpg)). Without action representation learning (left), the distinction between expert and suboptimal actions is not obvious. The latent actions of TRAIL (right), on the other hand, results in the expert latent actions being more visually separable from suboptimal actions.
>
> Note that qualitative evaluation of latent action space may not be reliable; one can probably tune the visualization to make any representation look "good", so we heavily rely on the quantitative results in our paper for testing latent action representations.
>
> > Typos in $\pi_k$ and $\pi_Z: S \to \Delta(Z)$
>
> Thank you for pointing out these typos; we have corrected them.
>
> > In Section 4.3 TRAIL EBM for Theorem 1, “we set $\rho$ to be the distribution of $s’$ in $d^{\text{off}}$”, does that mean you sample both the samples in the contrastive loss from the suboptimal data?
>
> Yes, both the positive sample $s'$ and negative sample $\tilde{s}'$ are sampled from the suboptimal data, but $s'$ is paired to $s$ (i.e., they belong to the same <$s, a, s’$> tuple, while $\tilde{s}'$ is independently sampled.

---

> > ### Author Response · Authors · 2021-11-19
> > **Author Follow-up**
> >
> > Dear Reviewer, we would like to ask if the continuous space sample complexity analysis and the additional visualization of the latent action space we have provided further strengthens our submission theoretically and qualitatively, or if there were any other issues that we can address. Please let us know, and thank you for your time.

---

> > > ### Comment · Reviewer_nwvX · 2021-11-21
> > > **Response to Author Response**
> > >
> > > I am sorry for the delay in the response.
> > >
> > > Thank you for providing helpful clarifications and adding the continuous action sample complexity analysis and latent action space visualization. They all make lots of sense.
> > >
> > > I have one remaining question. In the continuous action sample complexity analysis, the bound contains the C1 term in (1) which seems to still assume discrete actions (|A| appears in the C1 at the bottom of page 4). I wonder how that can be handled. Moreover, does the sample complexity proof require discrete state space? (I see |S| appears in the new bound provided and in THeorem 2.)

---

> > > > ### Author Response · Authors · 2021-11-22
> > > > **Clarification of Continuous Latent Action**
> > > >
> > > > Yes, our bounds still assume a discrete action space A. Note that the “continuous actions” in our sample complexity analysis refers to continuous latent actions (i.e., in the Z space) as opposed to continuous raw actions (i.e., in the A space). Avoiding discrete S or A for theoretical RL bounds involves more complex machinery and would have to include some sort of regularity assumption on the underlying MDP (e.g., Lipschitz [1]), which is out of the scope of our paper. We have updated the paper to make sure we always refer to “continuous latent actions” to avoid similar confusion.
> > > >
> > > > [1] https://link.springer.com/article/10.1007/s10994-015-5484-1

---

### Decision · Program_Chairs · 2022-01-20

**Decision:**

Accept (Poster)

**Comment:**

The paper investigates what we can learn from _suboptimal_ demonstrations for imitation learning. It suggests that we can learn about the structure of the environment by finding a factored dynamics model including a latent action space. It demonstrates both theoretically and empirically that this information can reduce sample requirements for downstream IL.

The reviewers praised the simplicity of the method (including its minimal assumptions), the theoretical analysis, and the breadth of the experimental validation. The authors were helpful during the discussion period, and addressed any questions or concerns the reviewers raised.

Overall, this is an interesting idea and a well-executed paper.